# Somatostatin-expressing parafacial neurons are $CO_2$/$H^+$ sensitive and regulate baseline breathing

**Colin M Cleary[1], Brenda M Milla[1], Fu-Shan Kuo[1], Shaun James[1], William F Flynn[2], Paul Robson[2,3], Daniel K Mulkey[1]***

[1]Department of Physiology and Neurobiology, University of Connecticut, Storrs, United States; [2]The Jackson Laboratory for Genomic Medicine, Farmington, United States; [3]Institute for Systems Genomics, University of Connecticut, Farmington, United States

**Abstract** Glutamatergic neurons in the retrotrapezoid nucleus (RTN) function as respiratory chemoreceptors by regulating breathing in response to tissue $CO_2$/$H^+$. The RTN and greater parafacial region may also function as a chemosensing network composed of $CO_2$/$H^+$-sensitive excitatory and inhibitory synaptic interactions. In the context of disease, we showed that loss of inhibitory neural activity in a mouse model of Dravet syndrome disinhibited RTN chemoreceptors and destabilized breathing (Kuo et al., 2019). Despite this, contributions of parafacial inhibitory neurons to control of breathing are unknown, and synaptic properties of RTN neurons have not been characterized. Here, we show the parafacial region contains a limited diversity of inhibitory neurons including somatostatin (*Sst*)-, parvalbumin (*Pvalb*)-, and cholecystokinin (*Cck*)-expressing neurons. Of these, Sst-expressing interneurons appear uniquely inhibited by $CO_2$/$H^+$. We also show RTN chemoreceptors receive inhibitory input that is withdrawn in a $CO_2$/$H^+$-dependent manner, and chemogenetic suppression of *Sst+* parafacial neurons, but not *Pvalb+* or *Cck+* neurons, increases baseline breathing. These results suggest *Sst*-expressing parafacial neurons contribute to RTN chemoreception and respiratory activity.

**\*For correspondence:**
daniel.mulkey@uconn.edu

**Competing interests:** The authors declare that no competing interests exist.

## Introduction

The retrotrapezoid nucleus (RTN) is an important respiratory control center located in the ventral parafacial region of the medulla oblongata (*Guyenet and Bayliss, 2015*; *Guyenet et al., 2019*). Of particular interest are RTN neurons that function as central respiratory chemoreceptors by regulating depth and frequency of breathing in response to changes in tissue $CO_2$/$H^+$ (*Guyenet et al., 2019*; *Nattie and Li, 2012*). Chemosensitive RTN neurons express the transcription factor Phox2b (*Stornetta et al., 2006*), are glutamatergic (*Mulkey et al., 2004*; *Weston et al., 2004*), and are intrinsically responsive to $CO_2$/$H^+$ by mechanisms involving acid inhibition of TWIK-related Acid-Sensitive $K^+$ channel 2 (TASK-2; *Wang et al., 2013a*) and activation of G-protein-coupled receptors (GPR4; *Kumar et al., 2015*). Activity of RTN chemoreceptors is also subject to modulation by purinergic signaling (*Mulkey et al., 2006*; *Wenker et al., 2012*) most likely from local astrocytes (*Gourine et al., 2010*; *Huckstepp et al., 2010*) as well as excitatory (*Lazarenko et al., 2011*; *Mulkey et al., 2007*) and inhibitory (*Guyenet et al., 2005*; *Moreira et al., 2007*; *Takakura et al., 2007*) synaptic input. In addition, previous work showed using a Phox2b-eGFP reporter line that RTN chemoreceptors have numerous presumptive inhibitory (symmetric) and glutamatergic synapses (asymmetric) (*Lazarenko et al., 2009*). These results are supported by functional evidence showing in urethane-anesthetized rats that unilateral RTN injection of glutamate receptor blockers (AP5 or CNQX) decreased baseline breathing and the ventilatory response to $CO_2$ (*Nattie and Li, 1995*),

whereas unilateral RTN injection of the GABA$_A$ receptor blocker bicuculline stimulated baseline breathing and blunted CO$_2$-stimulated respiratory activity (*Nattie et al., 2001*). This suggests glutamatergic and GABAergic synaptic drive to the RTN from distal and possibly local sources are required for maintenance of baseline breathing and CO$_2$/H$^+$ chemoreception. Furthermore, in the context of disease, suppression of parafacial inhibitory neural activity in a mouse model of Dravet syndrome caused a disordered breathing phenotype, and at the cellular level increased baseline activity and CO$_2$/H$^+$ dependent firing of RTN chemoreceptors (*Kuo et al., 2019*). These results suggest loss of inhibitory neural activity disinhibited RTN chemoreceptors and contributed to breathing problems. Despite this potential significance, contributions of parafacial inhibitory neurons to control of breathing are undetermined and synaptic properties of RTN neurons have not been characterized.

Based on in vivo multi-electrode array recordings in cats that showed the RTN contains CO$_2$/H$^+$-activated and -inhibited neurons that communicate through CO$_2$/H$^+$-dependent excitatory and inhibitory interactions (*Ott et al., 2011*), we hypothesize that ventral parafacial inhibitory neurons in the region of the RTN sense changes in CO$_2$/H$^+$ and regulate baseline breathing by a mechanism involving disinhibition. To address this, we first performed single cell RNA sequencing (scRNA-seq) to further characterize molecular signatures of chemosensitive RTN neurons and identify types of inhibitory neurons present in the parafacial region. We confirm the RTN is composed of two subsets of glutamatergic *Phox2b-* and *Nmb*-expressing neurons with similar levels of proton sensing machinery (*Gpr4* and *Kcnk5*) but differ in galanin (*Gal*) expression. We also determined that parafacial inhibitory neurons are predominantly GABAergic and glycinergic and are composed of *Sst*, *Pvalb*, *Ndnf*, and *Cck* subtypes, including an *Sst* subset that is strongly inhibited by CO$_2$/H$^+$. At the network level, we show that the RTN functions as a CO$_2$/H$^+$-sensing network where excitatory and inhibitory neurons interact in a CO$_2$/H$^+$-dependent manner to augment respiratory drive. Also, at the whole animal level, we show that ventral parafacial inhibitory neurons, and more specifically *Sst+* inhibitory neurons, regulate respiratory activity under baseline conditions but do not contribute to respiratory output under high CO$_2$ conditions. The CO$_2$/H$^+$ response profile of *Sst+* parafacial neurons, together with their preferential contribution to baseline breathing, suggest these cells are important determinants of resting respiratory drive.

## Results

### Molecular profiles of glutamatergic and GABA/glycinergic parafacial neurons

Single cells from the ventral parafacial region were isolated from 10-day-old wild type C57BL/6J (N = 16; 8 of each sex). Note that four male and four female mice included in this cohort received 4OH-tamoxifen (0.2 mg/daily for 3 days). However, since tamoxifen treatment did not affect the proportion of cells obtained or relative transcript profiles across all major cell types (*Figure 1—figure supplement 1*), these data sets were pooled. Single-cell RNA-seq was performed using the 10X Genomics Chromium Controller (*Zheng et al., 2017*) and 10X v2 chemistry targeting 16,000 barcoded cells. After quality control filtering and doublet removal (see Materials and methods), we analyzed 11,810 cells with a median of 2892 unique transcripts (UMIs) and 1472 genes per cell. We used the 2000 most highly variable genes, measured by dispersion, as input for dimensionality reduction using PCA, BBKNN, and UMAP, and found 20 distinct clusters using Leiden community detection (*Figure 1—figure supplement 1*).

Non-neuronal cells comprised roughly 90% of the dataset, so we employed a two-state mixture model parameterized on four general neuronal markers (*Snap25, Tubb3, Elavl2, Syp*) (*Mickelsen et al., 2019*) to segregate putative neurons from non-neuronal cells. A second mixture model was then used to classify the neurons into three groups: vesicular glutamate-transport type 2 (Vglut2+; *Slc17a6*) glutamatergic excitatory neurons (N = 197), vesicular GABA transporter (Vgat+; *Slc32a1*) inhibitory neurons (N = 445), and cholinergic (*Chat*) neurons (*Figure 1A*, *Figure 1—figure supplement 1*). The resulting neuronal populations had markedly higher molecular content, with a

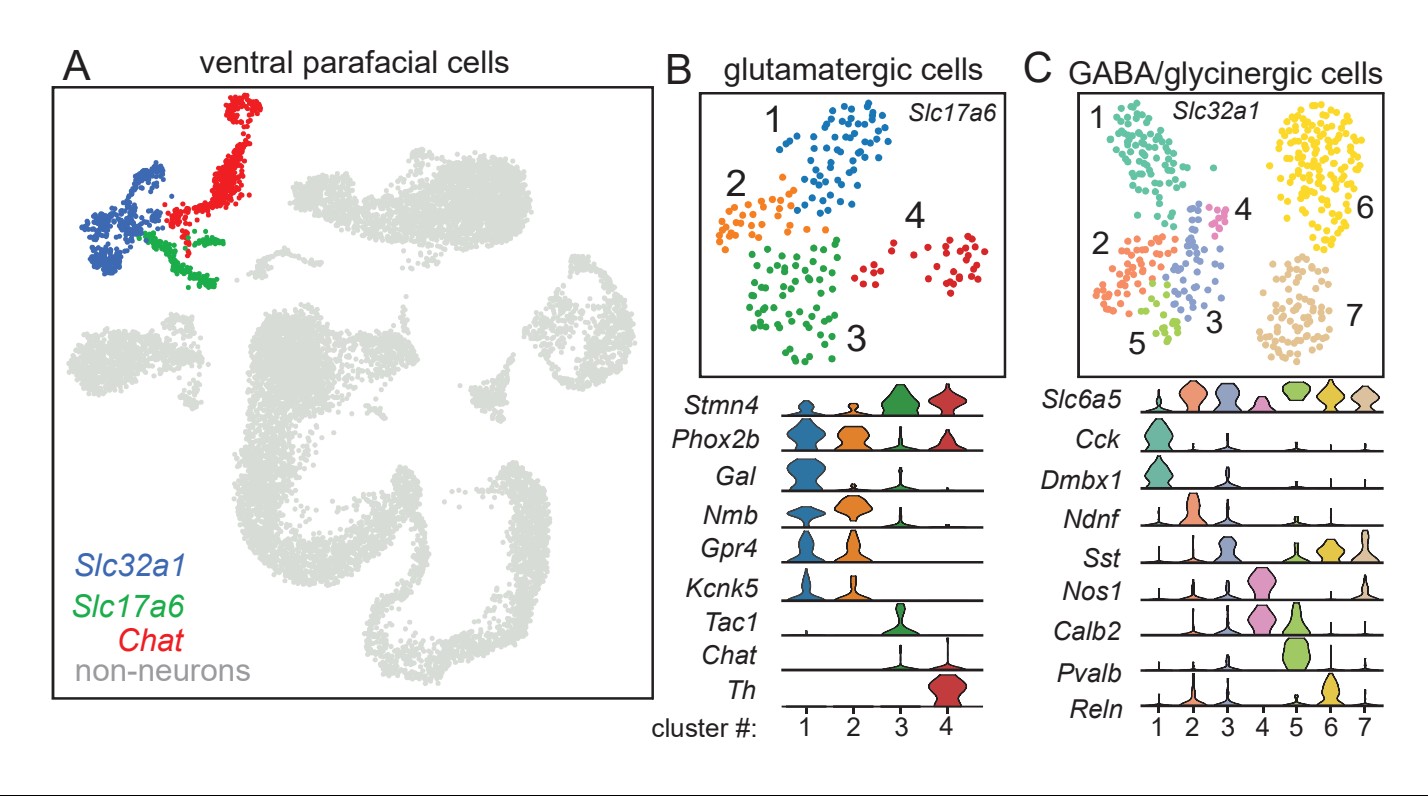

**Figure 1.** Molecular signatures of ventral parafacial glutamatergic and inhibitory neurons. (A) Through a normalized dispersion analysis for dimension reduction, a t-distributed stochastic neighbor embedding (t-SNE) was created of ventral parafacial single-cell transcriptome, with cells color coded by cluster. Neurons were differentiated from non-neurons (gray) based on expression of *Snap25*, *Syp*, *Tubb3*, and *Elavl2*. Cells expressing either *Slc17a6* or *Slc32a1* were used for sub-cluster analysis of glutamatergic and inhibitory neurons. A fairly large population of Chat expressing neurons was also detected but since neither glutamatergic (B) nor Vgat+ neurons express *Chat*, this population was not analyzed further. (B) UMAP plot depicting four sub-clusters of glutamatergic neurons and corresponding violin plots showing cluster-specific differential gene expression. Cluster number is noted on the x axis and gene expression (from 0 to 4 counts/cell) on the y axis. *Slc17a6* clusters 1–2 are presumed to be subsets of RTN chemoreceptors based on expression of *Phox2b*, *Nmb*, *Gpr4*, and *Kcnk5* that differ in expression of galanin. *Slc17a6* cluster 3 differentially expressed Tac1 (gene encoding substance P), suggesting that these cells may be parapyramidal raphe neurons, whereas cluster four differentially expressed tyrosine hydroxylase (Th) indicative of adrenergic C1 pre-sympathetic neurons. (C) UMAP plot showing seven sub-clusters of inhibitory (*Slc32a1*) neurons and corresponding violin plots showing cluster-specific differential gene expression. Cluster number is noted on the x axis and gene expression (from 0 to 4 counts/cell) on the y axis. Discrete subtypes of inhibitory neurons were identified based on non-overlapping expression of cholecystokinin (*Cck; Slc32a1* cluster 1), neuron-derived neurotrophic factor (*Ndnf, Slc32a1* cluster 2) and parvalbumin (*Pvalb; Slc32a1* cluster 5). Three somatostatin (*Sst*+) clusters could be differentiated based on expression of calretinin (*Calb2; Slc32a1* cluster 3), reelin (*Reln, Slc32a1* cluster 6) and neuronal nitric oxide synthase 1 (*Nos1*, Slc32a1 clusters 3, 7).

The online version of this article includes the following source data and figure supplement(s) for figure 1:

**Figure supplement 1.** Bioinformatic Pipeline used for single-cell RNA-seq analysis.

**Figure supplement 2.** Ventral parafacial inhibitory neurons are located in close proximity to RTN chemoreceptors.

**Figure supplement 2—source data 1.** Raw rostral to Caudal *Slc32a1* ^Cre ::TdT cell count.

**Figure supplement 3.** *Scn1a* transcript is expressed by sub-clusters of glutamtergic and GABA/glycinergic parafacial neurons.

median of 4587 UMIs and 2317 genes per *Slc17a6*+ neuron and 4439 UMIs and 2299 genes per *Slc32a1*+ neuron.

Two-dimensional embeddings and cluster assignments were generated for both the *Slc17a6*+ excitatory and *Slc32a1*+ inhibitory populations (*Figure 1B–C*) using a similar process described above (see Materials and methods). The population of *Slc17a6*+ cells does not overlap with *Chat*+ cells and includes two clusters (*Slc17a6* clusters 1–2) of *Phox2b*-positive and *Nmb* expressing neurons with similar levels of *Gpr4* (G-protein-coupled receptor 4) and *Kcnk5* (TASK-2; K2P5) but differ in galanin (*Gal*) expression (*Figure 1B*). The molecular profiles of these clusters are largely consistent with that of RTN chemoreceptors (*Shi et al., 2017*). Sympathetic C1 catecholamine neurons (*Slc17a6*

cluster 4) are identified by expression of tyrosine hydroxylase (*Th*) and *Phox2b* and the absence of *Nmb* (*Li et al., 2008*). The remaining glutamatergic cluster (*Slc17a6* cluster 3) was distinguished by expression of tachykinin 1 (*Tac1*, precursor for substance P), suggesting these cells are parapyramidal raphe neurons. All four glutamatergic clusters had detectable levels of stathmin 4 (*Stmn4*, encodes a microtubual binding protein [*Holmfeldt et al., 2003*]), suggesting these populations are not yet fully differentiated.

The population of parafacial inhibitory *Slc32a1*+ neurons was composed of 7 discrete clusters that could be distinguished by largely non-overlapping expression of somatostatin (*Sst*, *Slc32a1* clusters 3, 6–7), parvalbumin (*Pvalb*, *Slc32a1* cluster 5), cholecystokinin (*Cck*, *Slc32a1* cluster 1) and the more recently identified interneuron marker (*Abs et al., 2018*) neuron-derived neurotrophic factor (*Ndnf*, *Slc32a1* cluster 2) (*Figure 1C*). Of these, *Sst*+ interneurons were the most abundant cell type, but varied in expression of calretinin (*Calb2*, *Slc32a1* cluster 3), reelin (*Reln*, *Slc32a1* cluster 6) and neuronal nitric oxide synthase 1 (*Nos1*, *Slc32a1* clusters 3, 7). Consistent with other brainstem regions, we found that all *Slc32a1*+ parafacial neurons also express lysosome-associated membrane protein 5 (*Lamp5*) (*Koebis et al., 2019*) (not shown) and most express the sodium- and chloride-dependent glycine transporter 2 (GlyT2; *Slc6a5*) (*Hirrlinger et al., 2019*; *Figure 1C*), suggesting they have the capacity to release GABA and glycine. However, in contrast to the cortex (*Lee et al., 2010*; *Lim et al., 2018*), the ventral parafacial region was devoid of interneurons that express the serotonin receptor 5HT3aR (*Htr3a*), vasoactive intestinal peptide (*Vip*), or neuropeptide Y (*Npy*) (data not shown). In summary, these results provide the first molecular characterization of inhibitory neurons in a chemoreceptor region, and in doing so establish a cellular framework for understanding roles of inhibitory neurons in respiratory chemoreception.

We confirmed that TdT labeled *Slc32a1* cells are distributed throughout the ventral parafacial region including juxtaposed to Phox2b-immunoreactive RTN chemoreceptors (*Figure 1—figure supplement 2*). In-line with our scRNA-seq results, we found by fluorescent in situ hybridization that ~85% of *Slc32a1*+ cells (n = 57 cells) co-express glutamic acid decarboxylase (GAD67; *Gad1*) and GlyT2 (*Slc6a5*) (*Figure 1—figure supplement 2*). Also, consistent with our previous work showing that expression of a Dravet syndrome-associated *Scn1a* mutation disrupted RTN chemoreception and respiratory activity (*Kuo et al., 2016*), we found that *Scn1a* is expressed by all parafacial neurons, particularly inhibitory neurons that showed relatively higher levels of expression compared to glutamatergic neurons (*Figure 1—figure supplement 3*). Together, these results provide insight into the diversity of parafacial inhibitory neurons and identify unique molecular markers that may facilitate future assessment of cell type specific functions.

## *Sst*+ parafacial neurons are $CO_2/H^+$ sensitive

The function of inhibitory neurons in this region of the brainstem has not been characterized. However, they are located within a region that is specialized to sense changes in $CO_2/H^+$, therefore, we wondered whether parafacial inhibitory neurons also contribute to RTN chemoreception. These experiments were performed in slices isolated from a Cre recombinase-dependent reporter line (*Slc32a1^{Cre}*::TdT) which allows for selective targeting of inhibitory neurons in the region of interest (*Figure 2—figure supplement 1*). $CO_2/H^+$-sensitivity was characterized in cell-attached voltage-clamp mode; neurons that responded reversibly to 10% $CO_2$ (pH 7.0) with $\geq$20% change in activity were considered $CO_2/H^+$-sensitive. By this criterion, we found that exposure to 10% $CO_2$ decreased activity in 48 of 130 (37%) fluorescent parafacial neurons by an average of 2.0 ± 0.2 Hz (*Figure 2Ai, D*). The majority of parafacial inhibitory neurons (56%) did not respond to stimulus and so were considered $CO_2/H^+$-insensitive (*Figure 2Bi,D*), while a small subset of fluorescent parafacial neurons (7%) showed an excitatory response to $CO_2$ ($F_{2,130} = 91.23$, p<0.001) (*Figure 2C–D*). We also found the inhibitory effect of $CO_2/H^+$ on chemosensitive RTN neurons was retained when purinergic signaling was blocked by bath application of pyridoxalphosphate-6-azophenyl-2',4'-disulfonic acid (PPADS; 100 µM) and 8-phenyltheophylline (8-PT; 10 µM) ($T_7 = 0.1515$, p>0.05), or in the presence of ionotropic receptor blockers including CNQX (10 µM) to block AMPA/kainite receptors, gabazine (10 µM) to block $GABA_A$ receptors, and strychnine (2 µM) to block glycine receptors ($T_7 = 3.100$, p=0.02) (*Figure 2—figure supplement 2*). These results suggest parafacial inhibitory neurons are intrinsically $CO_2/H^+$-sensitive.

The next goal was to identify the molecular identity of $CO_2/H^+$-inhibited parafacial neurons. In other brain regions, subtypes of inhibitory neurons exhibit characteristic electrical properties and

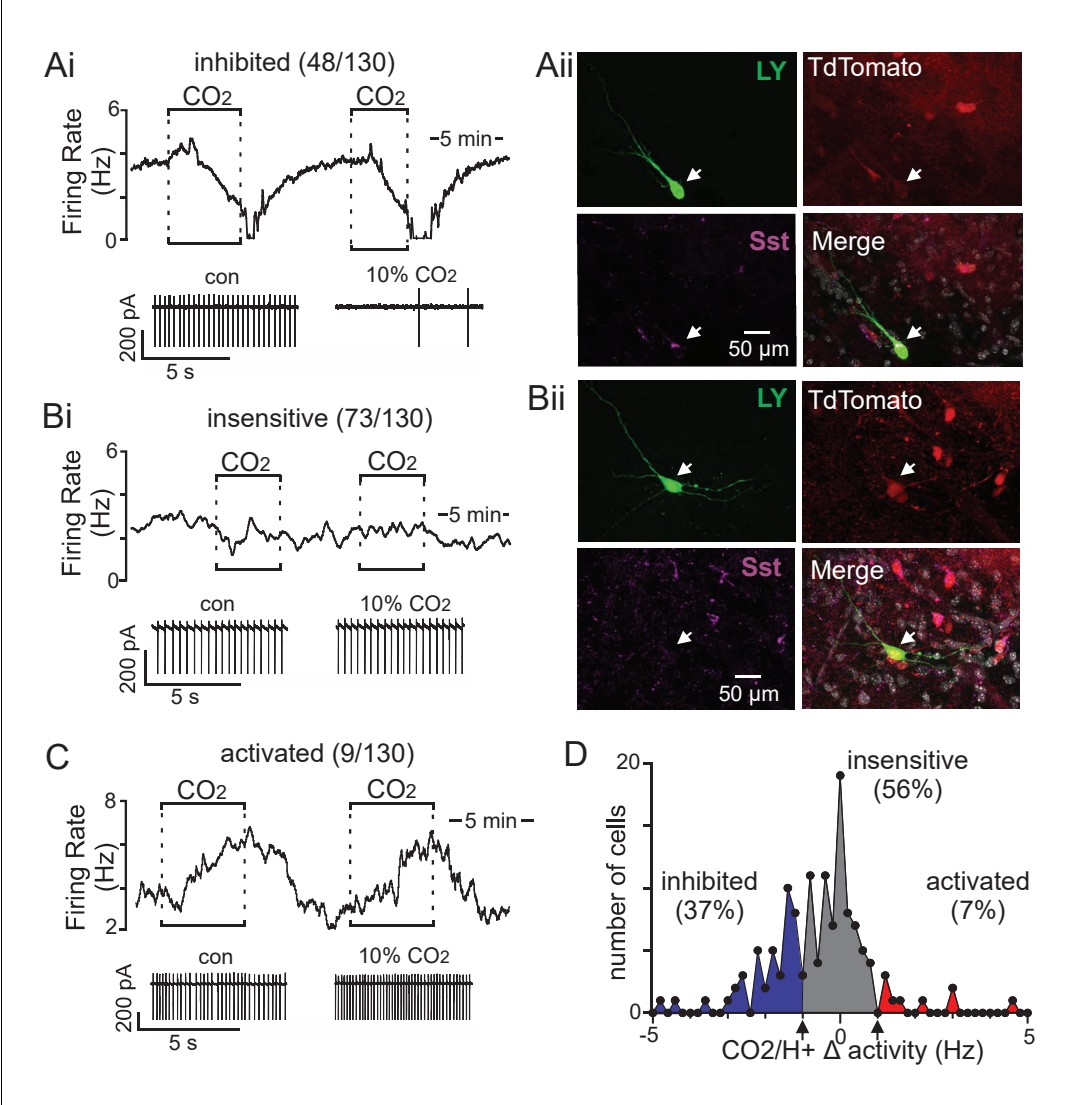

**Figure 2.** Somatostatin neurons in the ventral parafacial region are $CO_2$/$H^+$-sensitive. (A-C) Traces of firing rate and segments of holding current from ventral parafacial inhibitory neurons in slices from *Slc32a1*[Cre]::TdT mice show that exposure to 10% $CO_2$ suppressed activity by 2.0 ± 0.2 Hz in 43% of neurons tested (Ai), whereas the majority of *Slc32a1*+ neurons in this region did not respond (Δ firing 0.11 ± 0.04 Hz) to this same level of $CO_2$/$H^+$ (Bi), and a small minority are activated by $CO_2$/$H^+$ (C). (Aii) Example of a $CO_2$/$H^+$-inhibited Lucifer Yellow filled parafacial neuron that was Sst-immunoreactive (IR). (Bii) Example of a $CO_2$/$H^+$-*in*sensitive Lucifer Yellow filled parafacial neuron that was not Sst-IR. In sum, 5 of 5 $CO_2$/$H^+$-inhibited parafacial neurons were Sst-IR, whereas 0 of 5 $CO_2$/$H^+$-insensitive neurons in this region were Sst-IR. (D) Summary data (N = 130 neurons from 45 mice) plotted as number of cells vs mean firing response to 10% $CO_2$. Note that all neurons included in this analysis showed a similar baseline level of activity in 5% $CO_2$ (data not shown).

The online version of this article includes the following source data and figure supplement(s) for figure 2:

**Source data 1.** Parafacial inhibitory neuron responses to $CO_2$/$H^+$.
**Figure supplement 1.** Validation of the *Slc32a1*[Cre] mouse line.
**Figure supplement 2.** *Slc32a1*+ parafacial neurons are intrinsically inhibited by $CO_2$.

firing behavior (*Ascoli et al., 2008*). Although it is not clear whether such properties are discriminating for brainstem inhibitory neurons; nevertheless, in whole-cell current-clamp mode, we found $CO_2$/$H^+$-inhibited (662 MΩ) and -insensitive (703 MΩ) cells showed similar input resistances (p=0.91), baseline activity (p=0.59) and firing responses to depolarizing current injections (data not shown). These results suggest $CO_2$/$H^+$ sensitivity does not correlate with these electrical properties. Therefore, after gaining whole-cell access, we labeled cell types of interest with Lucifer Yellow (included in

the pipette internal solution) for *post hoc* immunohistochemical identification using markers based on results our single cell RNAseq analysis (*Figure 1C*). We found that 5 of 5 $CO_2/H^+$-inhibited cells were *Sst*-immunoreactive (*Figure 2Aii*) and were not immunoreactive for *Pvalb* or *Cck* (data not shown), whereas 0 of 5 $CO_2/H^+$-insensitive cells expressed Sst (*Figure 2Bii*). These results suggest $CO_2/H^+$-inhibited parafacial neurons are one or more types (clusters 3, 6–7; *Figure 1C*) of *Sst*-expressing inhibitory neurons.

## $CO_2/H^+$-synaptic properties of RTN chemoreceptors

Chemosensitive RTN neurons were identified in slices from *Slc32a1*[Cre]::TdT mice based on their lack of fluorescence and characteristic firing response to $CO_2/H^+$. As previously defined (*Kuo et al., 2016*), RTN neurons were considered chemosensitive if they show some level of spontaneous activity under control conditions and a robust firing rate response to 10% $CO_2$ ($\Delta 1.6 \pm 0.36$ Hz; N = 7 cells). Neurons that showed <1.0 Hz firing response to 10% $CO_2$ were considered non-chemosensitive and excluded from this study. Once the cell type of interest has been identified, we obtained whole-cell access and in voltage-clamp, recorded spontaneous synaptic currents. sIPSCs were recorded in relative isolation by holding cells at the reversal potential for AMPA-mediated EPSCs (sEPSCs; Ihold = 0 mV). Under control conditions (5% $CO_2$), chemosensitive RTN neurons showed sIPSCs with an

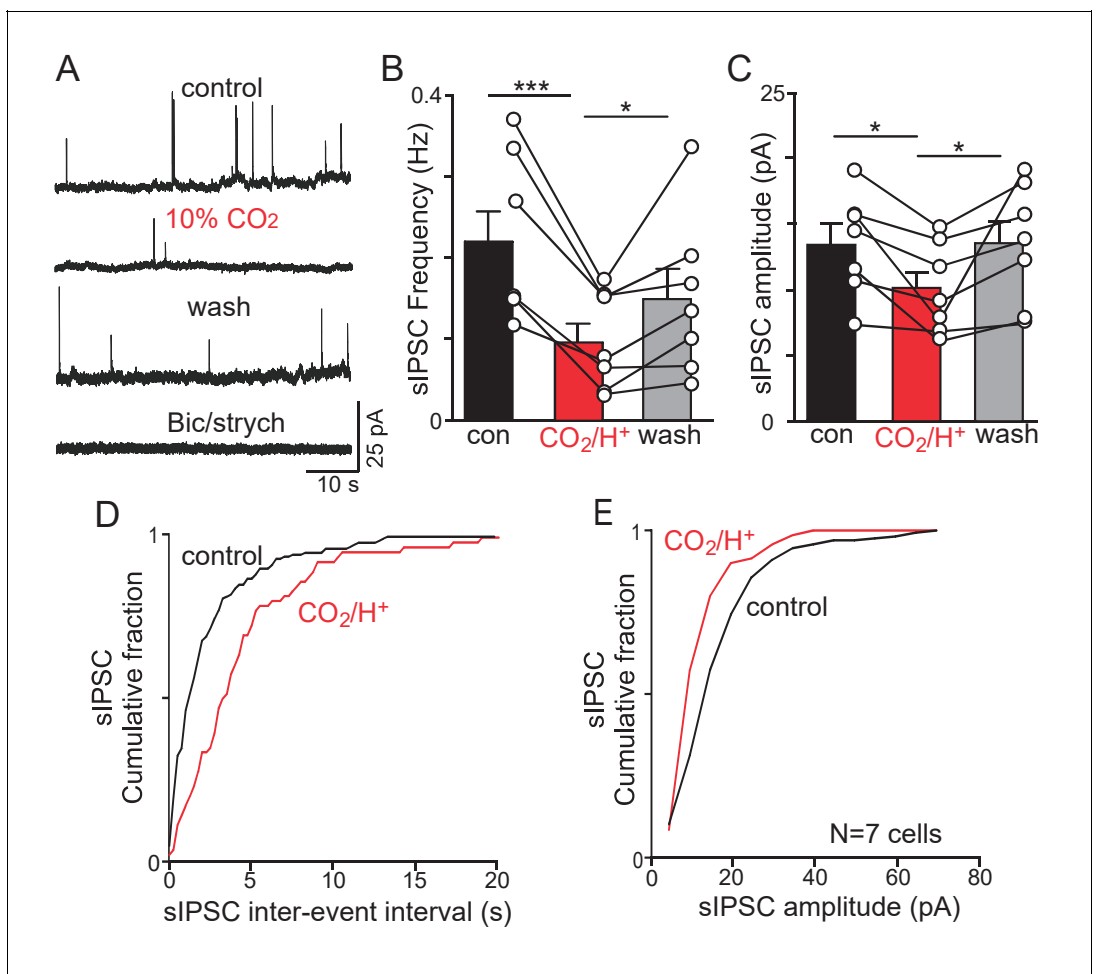

**Figure 3.** $CO_2/H^+$-dependent suppression of inhibitory synaptic input to RTN chemoreceptors. (**A**) Traces of holding current (Ihold = 0 mV) from an RTN chemoreceptor in a slice from a *Slc32a1*[Cre]::TdT mouse shows sIPSC events under control conditions and during exposure to 10% $CO_2$ or bicuculline (10 μM) and strychnine (2 μM). (**B–C**) Summary data (N = 7) show the average effect of $CO_2/H^+$ on sIPSC freq (**B**) and amplitude (**C**). (**D–E**) Effect of $CO_2/H^+$ on sIPSC frequency and amplitude are also reflected in cumulative distribution plots of sIPSC inter-event interval (D; bin size 250 ms) and amplitude (E; bin size 5 pA). Data was analyzed by one-way RM ANOVA followed by Tukey multiple comparison test. *p<0.05, **p<0.01 ***p<0.001.

average frequency of 0.22 Hz and amplitude of 13.7 pA (*Figure 3A–C*). The kinetics of sIPSCs recorded in RTN neurons are similar to what has been described in other brain regions (*Ali et al., 2007*; *Banks and Pearce, 2000*); average rise (10%–90%) and decay (90%–10%) times of 3.1 ms and 25.5 ms, respectively. Exposure to 10% $CO_2$ decreases sIPSC frequency to $0.1 \pm 0.06$ Hz ($F_{2,6}=21.04$; p<0.001; N = 7 cells total) (*Figure 3A–B*) which corresponded with an increase in the inter-event interval (*Figure 3D*) and decreased sIPSC amplitude ($F_{2,6}=6.58$; p<0.05) (*Figure 3C*) but with no change in sIPSC rise (3.7 msl p=0.516) or decay (27.7 ms; p=0.268) times. The amplitude and frequency of sIPSCs returned to near control levels after washing back to 5% $CO_2$ (*Figure 3B–C*). Subsequent bath application of bicuculline (10 µM) to block $GABA_A$ receptors and strychnine (2 µM) to block glycine receptors eliminated all sIPSCs, thus confirming they are mediated by GABA or glycinergic input (*Figure 3A*). Although we cannot exclude the possibility that $CO_2/H^+$ disrupts transmitter release from synaptic terminals in the RTN from distal inhibitory neurons, nevertheless, these results are consistent with the possibility that a subset of parafacial *Slc32a1* neurons are inhibited by $CO_2/H^+$ and contribute to RTN chemoreception by a mechanism involving disinhibition.

To further test this possibility, we characterized baseline activity and $CO_2/H^+$-sensitivity of RTN chemoreceptors under control conditions and when GABA and glycine transmission was blocked with bicuculine and strychnine. In cell-attached voltage-clamp mode, bath application of bicuculine (10 µM) and strychnine (2 µM) increased baseline activity of RTN chemoreceptors by $0.7 \pm 0.2$ Hz ($T_{12} = 2.201$, p=0.022) (data not shown). This finding suggests inhibitory input partly limits activity of RTN chemoreceptors under baseline conditions. We also found the firing response to 10% $CO_2$ was similar under control conditions and in the presence of bicuculine and strychnine ($\Delta -0.3 \pm 0.2$ Hz; $T_{12} = 1.246$, p>0.05) (data not shown). These results suggest $CO_2/H^+$-dependent inhibitory transmission regulates activity of RTN chemoreceptors under baseline conditions but not during exposure to high $CO_2$.

Since application of glutamate receptor blockers into the RTN blunted the ventilatory response to $CO_2$ (47), we next characterized $CO_2/H^+$-dependent modulation of sEPSC's in chemosensitive RTN neurons. Under control conditions (5% $CO_2$) and at a holding potential of −60 mV chemosensitive RTN neurons exhibit sEPSCs with an average frequency of 0.33 Hz and amplitude of −12.6 pA (*Figure 4A–C*). The kinetics of RTN sEPSCs are similar to AMPA-mediated events described in other brain regions (*Magee and Cook, 2000*; *Rodriguez-Molina et al., 2007*; *Selyanko et al., 1979*); average rise (10%–90%) and decay (90%–10%) times of 1.2 ms and 4.9 ms, respectively. This was also confirmed pharmacologically by bath application of CNQX (10 µM) at the end of each experiment (*Figure 4A*). Exposure to 10% $CO_2$ increased sEPSC frequency to $0.65 \pm 0.08$ Hz ($F_{2,7} = 26.91$; p<0.001) (*Figure 4A–B*) which corresponded with a decrease in the inter-event interval (*Figure 4D*) but with no change in sEPSC amplitude ($F_{2,7} = 1.398$; p>0.05) (*Figure 4C*) or kinetics (rise time 1.3 ms, p=0.5772; decay time 5.0 ms, p=0.9642). Furthermore, when both excitatory and inhibitory spontaneous synaptic events were characterized in the same cells (N = 6), we confirmed that $CO_2/H^+$ increases the sEPSC/sIPSC ratio from 1.5 to 8.9 (*Figure 4F*) ($T_5 = 3.70$; p<0.05). The differential effects of $CO_2/H^+$ on excitatory and inhibitory synaptic currents also argues against potential non-specific effects of $H^+$ on neurotransmission (*Sinning and Hübner, 2013*). Together, these results support the possibility that the RTN functions as a $CO_2/H^+$-sensing network where excitatory and inhibitory neurons interact in a $CO_2/H^+$-dependent manner to augment respiratory drive.

## *Sst+* parafacial neurons regulate baseline breathing

Evidence suggests *Sst*-expressing parafacial neurons are inhibited by $CO_2/H^+$ (*Figure 2*) and contribute to basal activity of RTN chemoreceptors. To test this possibility in vivo, we used an AAV delivery system to express an inhibitory (Gi-coupled) DREADD receptor in a Cre-recombinase-dependent manner in *Sst+* neurons. Specifically, we injected AAV2-hSyn-DIO-hM4D(Gi)-mCherry (10 nL/side, Addgene) bilaterally into the medial parafacial region of *Sst*^Cre mice (JAX #: 013044) (*Figure 5*). The virus spread laterally to also include a lateral portion of the parafacial region associated with active expiration (*Figure 5—figure supplement 1*). After 2 weeks recovery, we characterized baseline breathing and the $CO_2$ ventilatory response following sequential injections (I.P.) of saline followed ~2.5 hr later by clozapine (1 mg/kg; I.P.). These experiments began under room air conditions followed by exposure to 0, 3, 5, and 7% $CO_2$ (balance $O_2$ to limit peripheral chemoreceptor input). Animals were exposed to each condition for 10 min and each trial was limited to a total duration of 50 min to minimize the impact of clozapine clearance on receptor activation (*Jendryka et al.,*

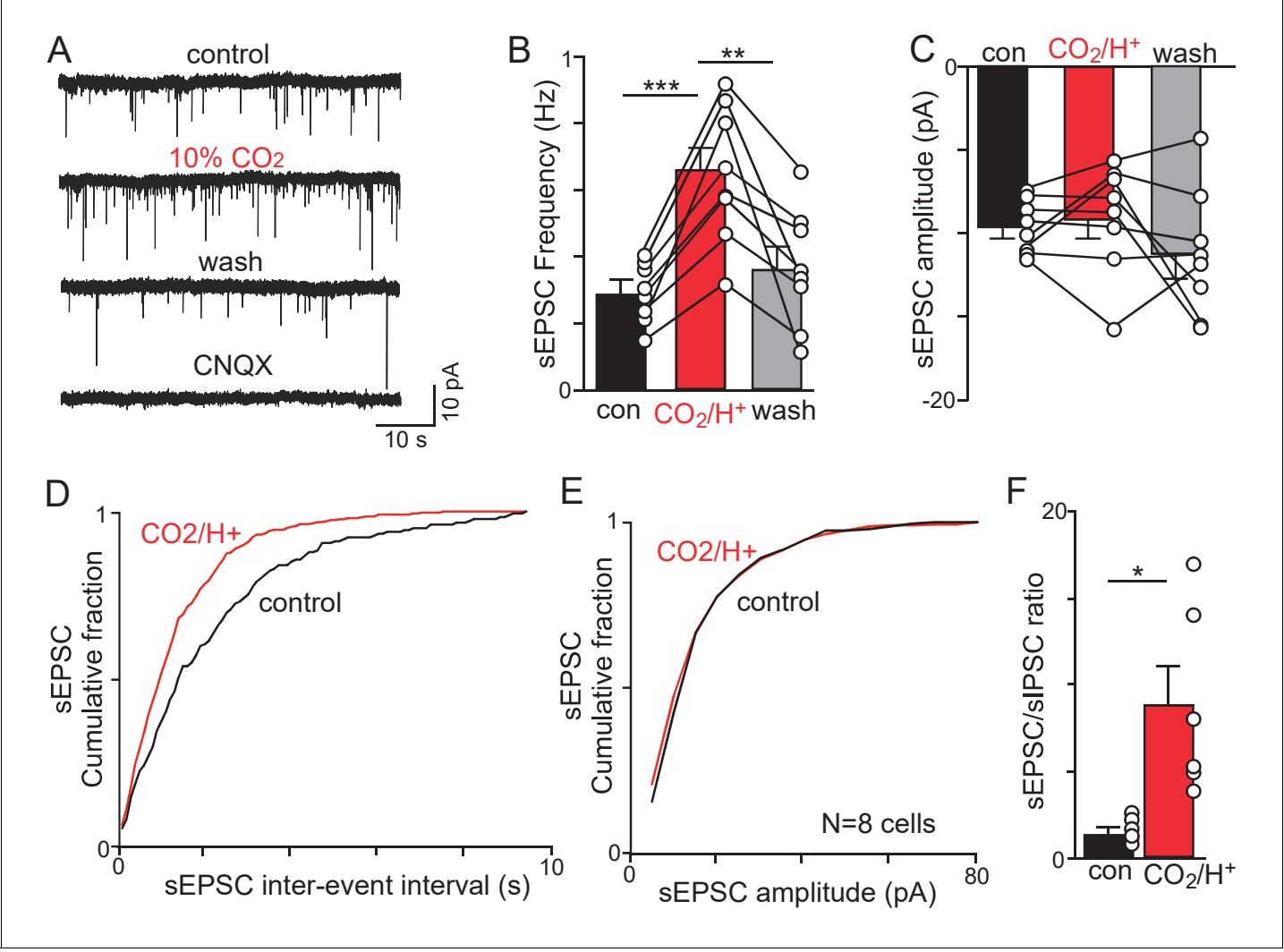

**Figure 4.** $CO_2/H^+$-dependent facilitation of excitatory glutamatergic input to RTN chemoreceptors. (**A**) Traces of holding current (Ihold = −60 mV) from a chemosensitive RTN neuron in a slice from a *Slc32a1*[Cre]::TdT mouse shows sEPSC events under control conditions and during exposure to 10% $CO_2$ or CNQX (10 µM). (**B-C**) summary data (N = 8) show the average effect of $CO_2/H^+$ on sEPSC freq (**B**) and amplitude (**C**). Note that two outlier data points were excluded from analysis. (**D–E**) Effect (or lack thereof) of $CO_2/H^+$ on sEPSC frequency and amplitude are also shown in cumulative distribution plots of sIPSC inter-event interval (D; bin size 100 ms) and amplitude (E; bin size 5 pA). (**F**) $CO_2/H^+$-induced suppression of sIPSC frequency in conjunction with increased sEPSC frequency resulted in enhancement of the sEPSC/sIPSC ratio. *p<0.05, **p<0.01 ***p<0.001.

*2019*). Consistent with our cellular (*Figure 2*) and synaptic data (*Figures 3–4*), we found that chemogenetic suppression of *Sst*+ parafacial neurons increased respiratory frequency ($F_{1,5}$=148.1, p<0.0001) (*Figure 5A–B*), tidal volume ($F_{1,5}$ = 7.360, p=0.0421) (*Figure 5C*) and minute ventilation ($F_{1,5}$ = 81.06, p=0.0003) under baseline conditions (*Figure 5D*). However, clozapine and saline treated mice showed similar ventilatory responses to $CO_2$ ($F_{1,5}$ = 0.9089, p>0.05) (*Figure 5D*). This is not surprising since $CO_2/H^+$-dependent suppression of inhibitory neural activity may preclude the effects of further chemogenetic inhibition on respiratory activity. Interestingly, parallel experiments performed in *Pvalb*[Cre] (JAX #: 008069) ($F_{1,5}$ = 0.0139, p>0.05) and *Cck*[Cre] (JAX #: 012706) ($F_{1,5}$ = 0.0660, p>0.05) minimally affected baseline breathing or the ventilatory response to $CO_2$ (*Figure 5E–F*). It should be noted that cell type-specific expression of Cre recombinase in *Sst*[Cre] (*Soumier and Sibille, 2014*), *Pvalb*[Cre] (*Liu et al., 2019*) and *Cck*[Cre] (*Matsuda et al., 2020*) lines have been confirmed previously. Together with our cellular evidence, these results suggest *Sst*+ parafacial neurons are specialized to sense changes in $CO_2/H^+$ and contribute to respiratory activity.

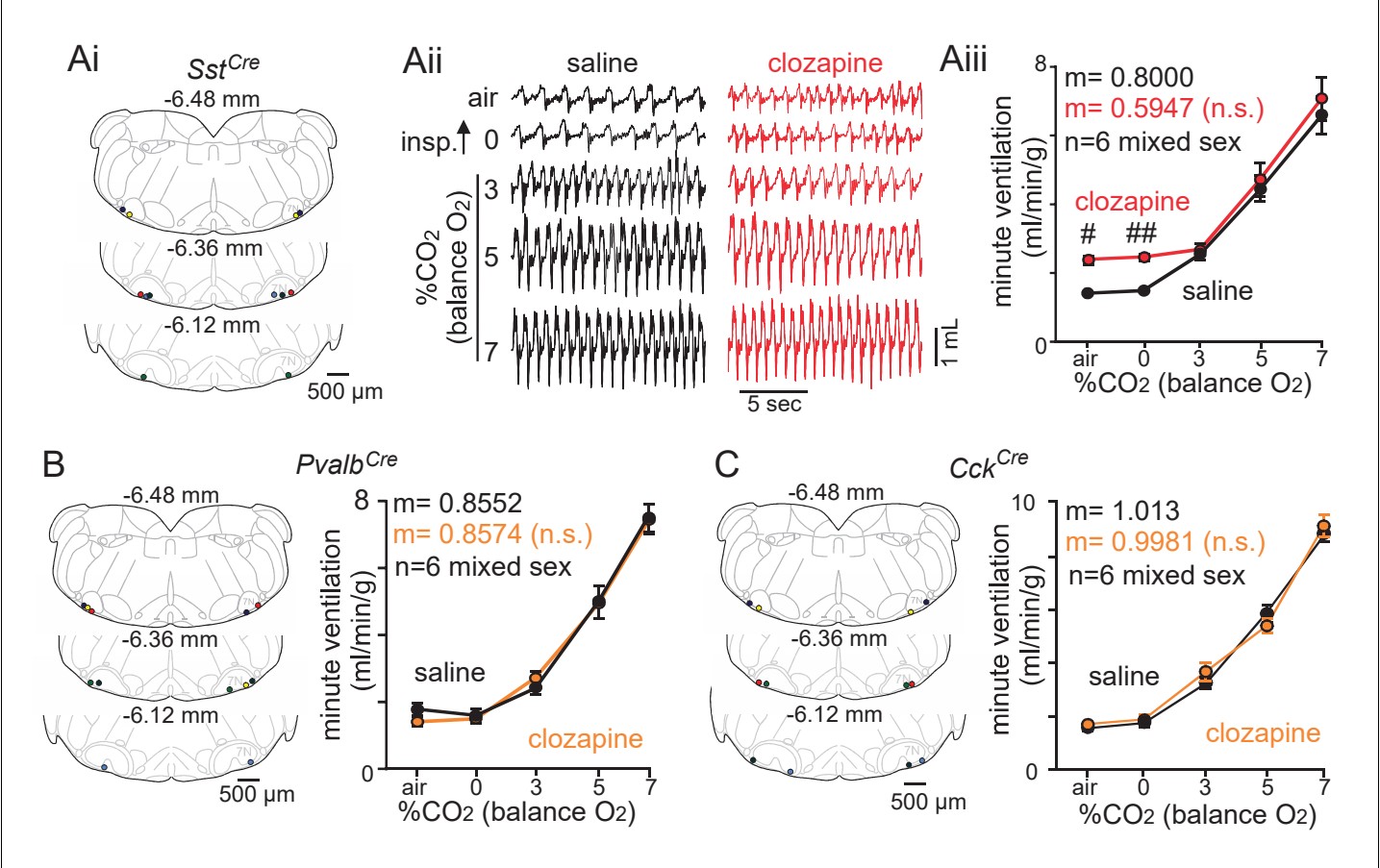

**Figure 5.** Chemogenetic suppression of *Sst*+ but not *Pvalb*+ or *Cck*+ parafacial neurons increased baseline breathing. (**Ai**) Computer-assisted plots show centers of bilateral AAV2-hSyn-DIO-hM4D-mCherry injections in *Sst*$^{Cre}$ animals. The number above each section indicates the relative position of each slice behind bregma (***Paxinos and Franklin, 2013***). (**Aii**) Traces of respiratory activity from *Sst*$^{Cre}$ mice that received bilateral parafacial injections of AAV2-hSyn-DIO-hM4D-mCherry following systemic (I.P.) injection of clozapine (1 mg/kg) or saline (control). Aiii, summary data (n = 6) shows effects of chemogenetic suppression Sst+ parafacial neurons with clozapine on minute ventilation under room air conditions and in 0–7% $CO_2$ (balance $O_2$). (**B–C**) Left side shows computer-assisted plots of AAV2-hSyn-DIO-hM4D-mCherry injection centers in *Pvalb*$^{Cre}$ (**B**) and *Cck*$^{Cre}$ (**C**) mice, and summary data to the right shows effects of chemogenetic suppression of parafacial *Pvalb*$^+$ (**B**) and *Cck*$^+$ (**C**) neurons on minute ventilation under room air conditions and in 0–7% $CO_2$ (balance $O_2$). N = 6 for each genotype (mixed sex). Slopes between 0% and 7% $CO_2$ were compared by analysis of covariance (ANCOVA). $^\#$, different between genotypes (two-way ANOVA with Tukey's multiple comparison test; one symbol p<0.05, two symbols p<0.01).

The online version of this article includes the following figure supplement(s) for figure 5:

**Figure supplement 1.** Chemogenetic suppression of all ventral parafacial neurons increased baseline breathing.

**Figure supplement 2.** Clozapine alone minimally effects baseline breathing or the chemoreflex.

We performed similar experiments in *Slc32a1*$^{Cre}$ mice (JAX #: 016962) to suppress activity of all parafacial inhibitory neurons including clusters 2 and 4 from our scRNAseq dataset (***Figure 1C***), which were not targeted in experiments described above. We found that chemogenetic suppression of all parafacial inhibitory neurons mirrored the effects of targeted inhibition of just *Sst*+ neurons. For example, *Slc32a1*$^{Cre}$ mice responded to clozapine (1 mg/kg) with an increase in baseline respiratory frequency ($F_{1,9}$=5.560, p=0.043) and tidal volume ($F_{1,9}$ = 19.13, p=0.002), which culminated in an increase minute ventilation ($F_{1,9}$=6.017, p=0.037) (***Figure 5—figure supplement 1***). Also, as observed in *Sst*$^{Cre}$ mice, we found that inhibition of all parafacial inhibitory neurons by clozapine administration (I.P.) in Slc32a1$^{Cre}$ mice minimally effected the $CO_2$ chemoreflex ($F_{1,9}$=0.0724, p>0.05) (***Figure 5—figure supplement 1***). We confirmed Cre recombinase is specific to inhibitory parafacial neurons in Slc32a1$^{Cre}$ mice (95% of TdT fluorescence co-localized with Gad67-immunoreactivity), which confirms previously reported data (***Lowery-Gionta et al., 2018***; ***Figure 5—figure supplement 1***). It should also be noted that clozapine had minimal effect on respiratory activity in

Slc*32a1*$^{Cre}$ mice that did not receive intracranial viral injections ($F_{1,5}=0.2399$, p>0.05) (*Figure 5—figure supplement 2*). Together, these results identify *Sst*-expressing parafacial neurons as important determinants of baseline breathing.

## Discussion

This study provides the first characterization of synaptic properties of RTN neurons and in doing so identifies a role of ventral parafacial inhibitory neurons in $CO_2/H^+$ dependent control of breathing. Specifically, we show that (1) the ventral parafacial region contains *Sst*, *Ndnf*, *Pvalb*, and *Cck* classes of interneurons of which only *Sst*-expressing neurons are inhibited by $CO_2/H^+$; (2) chemosensitive RTN neurons receive inhibitory input under control conditions that is withdrawn during exposure to high $CO_2$; (3) chemogenetic inhibition of *Sst*+ but not *Pvalb*+ or *Cck*+ parafacial neurons increases baseline respiratory activity. These results suggest *Sst*-expressing parafacial neurons are important determinants of respiratory activity under baseline conditions. This is important because disruption of baseline breathing is the root cause of disordered breathing in various disease states.

### Molecular profile of glutamatergic and GABA/glycinergic ventral parafacial neurons

Consistent with previous work (*Dubreuil et al., 2009*; *Shi et al., 2017*; *Stornetta et al., 2006*), we show that several types of ventral parafacial neurons express *Phox2b* including presumptive cholinergic neurons (*Chat*+), C1 pre-sympathetic neurons (*Th*+) and two subsets that express *Nmb* and similar levels of $H^+$ sensing machinery (*Gpr4* and *Kcnk5*) but differ in galanin expression. It is not clear whether both *Nmb*+ populations have similar or divergent roles in control of breathing. Based on *Gpr4* and *Kcnk5* expression both *Nmb*+ subtypes likely function as chemoreceptors, however, $CO_2/H^+$-activated *Phox2b*+ parafacial neurons also contribute to pre-inspiratory rhythmogenesis early in development (*Onimaru et al., 2008*) so perhaps differential expression of galanin denotes these functional differences. It is also not clear whether expiratory parafacial neurons express *Phox2b* and respond to $CO_2/H^+$ similar to RTN chemoreceptors or comprise a functionally discrete respiratory center. For example, selective activation (*Souza et al., 2020*) or inhibition (*Marina et al., 2010*) of ventral parafacial *Phox2b*-expressing neurons increased and decreased expiratory activity, respectively, suggesting expiratory parafacial neurons may be an extension of *Phox2b*-positive RTN chemoreceptors that differentially regulate inspiratory or expiratory activity depending on projection targets. However, others have shown that expiratory parafacial neurons are not Phox2b-immunoreactive (*de Britto and Moraes, 2017*), suggesting these neurons are distinct from RTN chemoreceptors. In any case, since expiratory parafacial neurons are putatively glutamatergic (*Silva et al., 2016*), this population is most likely included in clusters 1–3. Of these, cluster 3 is of particular interest because it lacks *Gpr4* and *Kcnk5* and so is not likely to function as an RTN chemoreceptor. It is also worth noting that *Slc17a6* clusters 3–4 express high levels of *Stmn4*, an important regulator of neural differentiation (*Lin and Lee, 2016*), so perhaps these populations diverge at later developmental time points. In any case, we are unable to disentangle putative rhythmogenic or expiratory parafacial neurons from other *Phox2b*+ cell types at this time. Nonetheless, it is also important to recognize that *Phox2b* is expressed by multiple cell types in the parafacial region, thus confounding interpretation of previous work that relied primarily on *Phox2b* expression to target RTN chemoreceptors (*Gourine et al., 2010*).

We also determined that the ventral parafacial region contains a limited diversity of inhibitory neurons including one *Cck*+ cluster, one *Pvalb*+ cluster, one *Ndnf*+ cluster and three *Sst*+ clusters that differ in terms of calretinin, reelin and *Nos1* expression. Although our cellular experiments suggest Sst+ cells are $CO_2/H^+$-sensitive (*Figure 2*), we were not able to identify which Sst+ cluster(s) function as chemoreceptors based on gene expression. We also found the ventral parafacial region did not include *Htra3*-, *Vip*-, or *Npy*-expressing inhibitory neurons. This is in marked contrast to the cortex where *Htra3*+ inhibitory neurons represent ~30% of the total interneuron population (*Tremblay et al., 2016*). By defining the diversity if inhibitory neurons in this brainstem region, these results provide a basis for understanding how disruption of inhibitory neural function in diseases like Dravet syndrome cause brainstem disfunction and mortality.

### Sst-expressing parafacial neurons are intrinsically inhibited by $CO_2$

The RTN is an important respiratory chemoreceptor region (*Guyenet et al., 2019*); therefore, it is reasonable to speculate that any neurons in this region that respond to $CO_2/H^+$ might do so in a manner tailored to support this reflex. Consistent with this, we found that a subset of inhibitory parafacial neurons (37%) are inhibited by 10% $CO_2$. This response was retained when purinergic signaling or fast neurotransmission was blocked, suggesting these cells are intrinsically $CO_2/H^+$ sensitive. We also showed that $CO_2/H^+$-inhibited parafacial neurons express *Sst* while $CO_2/H^+$-insensitive cells did not, thus honing the list of inhibitory chemoreceptor candidates to clusters 3, 6, and 7 (*Figure 1C*). Since the majority of inhibitory neurons in this region did not respond to this level of $CO_2$, it is tempting to speculate that *Sst+* parafacial neurons are specialized to contribute to RTN chemoreceptors and respiratory drive.

## The network basis of RTN chemoreception

Previous evidence based on multi-electrode extracellular recordings from cats suggests that RTN neurons including $CO_2/H^+$-activated (presumably chemoreceptors) and -inhibited (most likely *Slc32a1+* based on evidence presented in this study) interact through paucisynaptic connections (*Ott et al., 2011*). Consistent with this, we show that exposure to high $CO_2/H^+$ increased the frequency of spontaneous glutamatergic input to RTN neurons while simultaneously decreasing frequency and amplitude of spontaneous inhibitory synaptic inputs.

Mechanisms contributing to $CO_2/H^+$-induced activation of the RTN network likely involves both pre- and post-synaptic mechanisms. At the presynaptic level, we consider other chemosensitive RTN neurons or medullary raphe neurons which may co-release glutamate (*El Mestikawy et al., 2011*) as most the likely sources of $CO_2/H^+$ dependent glutamatergic drive to RTN chemoreceptors. Also, since $CO_2/H^+$ inhibited the activity of some parafacial *Slc32a1+* neurons, we consider these cells the most likely substrate responsible for $CO_2/H^+$-induced suppression of sIPSC frequency. It should be noted that few inhibitory parafacial neurons were activated by $CO_2/H^+$ (*Figure 3C*), suggesting RTN chemoreceptors do not project to and regulate activity of parafacial inhibitory neurons. The differential effects of $CO_2/H^+$ on EPSC and IPSC frequency also argue against potential non-specific inhibitory effects of $CO_2/H^+$ on voltage gated $Ca^{2+}$ channels (*Shah et al., 2001*) and neurotransmission (*Sinning and Hübner, 2013*). However, the RTN (*Guyenet et al., 2005*; *Takakura et al., 2007*; *Yang and Feldman, 2018*) and parafacial region (*Silva et al., 2020*) receive inhibitory input from various elements of the respiratory circuit that serves to limit chemoreceptor function during times of high respiratory activity, and we cannot exclude potential effects of $CO_2/H^+$ on transmitter release from these inputs.

At the postsynaptic level, our finding that $CO_2/H^+$ minimally affected sEPSC amplitude is consistent with evidence that AMPA receptors are largely unaffected by acidification. Conversely, $CO_2/H^+$-dependent suppression of sIPSC amplitude may involve $H^+$-dependent inhibition of GABA or glycine receptors on RTN neurons. For example, depending on the subunit composition, certain recombinant $GABA_A$ receptors (*Huang et al., 2004*; *Huang and Dillon, 1999*; *Wang et al., 2005*) and glycine $\alpha 1$ and $\alpha 1\beta$ receptors (*Chen et al., 2004*) are reversibly inhibited by acidification near the physiological range. It should also be noted that $CO_2/H^+$ may elicit release of non-glutamatergic neuromodulators that alter postsynaptic conductance and potentially contribute to diminished IPSC amplitude; however, we consider this unlikely since this non-specific mechanism is expected to affect both excitatory and inhibitory synaptic currents. Together these results suggest the RTN functions as a $CO_2/H^+$-sensing network composed of both faciliatory and disinhibitory interactions.

## Parafacial inhibitory neurons contribute to baseline breathing

The possibility that disinhibition contributes to chemoreception is not novel or unique to the RTN. For example, $CO_2/H^+$-inhibited cells have been found in several putative chemosensitive regions (*Conrad et al., 2009*; *Nichols et al., 2009*; *Wang and Richerson, 1999*) including GABAergic neurons in the medullary raphe (*Iceman et al., 2014*) and parafacial region (*Kuribayashi et al., 2008*). These results suggest disinhibition contributes to respiratory chemoreception. However, in the absence of evidence that inhibitory neurons in these regions actually influence chemoreceptor function or contribute to respiratory behavior, this possibility has remained largely speculative. We addressed these knowledge gaps by first showing that chemosensitive RTN neurons receive

inhibitory synaptic input under control conditions that is withdrawn in a $CO_2/H^+$-dependent manner (*Figure 4*). We also show that chemogenetic suppression of *Sst+* parafacial neurons increased minute ventilation (frequency and tidal volume) (*Figure 5A*). Therefore, we propose that *Sst+* parafacial neurons contribute to the drive to breathe by regulating baseline activity of RTN chemoreceptors. When the activity of *Sst+* parafacial neurons is diminished under high $CO_2$ conditions, it is perhaps not surprising that further inhibition of this population by chemogenetic means had negligible effect on respiratory output. These results are consistent with pharmacological evidence from anesthetized rats showing that application of bicuculline ($GABA_A$ receptor blocker) to the ventrolateral medulla increased inspiratory activity under control conditions but not during exposure to high $CO_2$ (*Gourine and Spyer, 2001*). Furthermore, disinhibition of neurons located more laterally in the ventral parafacial region contributed to emergence of active expiration during exposure to high $CO_2$ (*Huckstepp et al., 2015*; *Pagliardini et al., 2011*). Recent work suggests GABAergic neurons in medullary raphe regions regulate expiratory output of the lateral parafacial region during high $CO_2$ (60); however, the source of $CO_2/H^+$-dependent disinhibition of expiratory parafacial neurons remains unclear. Our evidence that *Sst+* parafacial neurons show a $CO_2/H^+$ response consistent with disinhibition and their close proximity to the lateral parafacial region makes them ideal candidates for such a function. However, potential roles of these neurons in regulation of expiratory activity requires further investigation.

### Potential physiological significance

The ability of respiratory chemoreceptors to sense and respond to tissue $CO_2/H^+$ is what maintains breathing during sleep. As $CO_2/H^+$ levels decrease so too does respiratory activity until the apneic threshold is reached and breathing ceases. Therefore, baseline $CO_2$ levels and the apneic threshold are critical determinants of stable breathing during sleep (*Nakayama et al., 2002*). The ability of ventral parafacial Sst+ neurons to respond to $CO_2$ in the low physiological range (minimally active under high $CO_2$ conditions) and preferentially regulate baseline breathing suggests these cells are important determinants of baseline $CO_2$ levels and the apneic threshold. Although we did not observe an apnea phenotype during chemogenetic suppression of parafacial inhibitory neurons, these experiments were performed in awake mice in which arousal-dependent mechanism may help stabilize breathing. Interestingly, a yet unidentified population of *Slc32a1+* neurons in nearby and potentially overlapping medullary regions are known to regulate rapid eye movement (REM) sleep, where activation of these cells promotes a REM-like state and inhibition of these cells does the opposite (*Weber et al., 2015*). Considering that $CO_2/H^+$ also stimulates arousal (*Guyenet and Bayliss, 2015*), it is possible $CO_2/H^+$ inhibition of parafacial *Slc32a1+* neurons suppresses REM and promotes arousal, thus directly coordinating chemoreceptor activity with sleep-wake state. However, these possibilities require further investigation. An additional caveat to note is that in addition to RTN chemoreceptors, *Sst+* parafacial neurons likely regulate respiratory activity at other levels of the respiratory circuit. Therefore, an important future direction of this work will be to identify projection targets of *Sst+* parafacial neurons.

# Materials and methods

**Key resources table**

| Reagent type (species) or resource | Designation | Source or reference | Identifiers | Additional information |
|---|---|---|---|---|
| Strain, strain background (*M. musculus*, Vgat-iris-Cre, mixed 129/SvJ and C57BL6/J background) | Slc32a1tm2(cre)Lowl/J | Jackson Laboratories | RRID:IMSR_JAX:016962 | |
| Strain, strain background (TdTomato reporter Ai14, C57BL6/J background) | B6.Cg-Gt(ROSA)26Sor^tm9 (CAG-tdTomato)Hze/J | Jackson Laboratories | RRID:IMSR_JAX:007909 | |
| Strain, strain background (Phox2b Cre, C57Bl6/J background) | B6(Cg)-Tg(Phox2b-cre)3Jke/J | Jackson Laboratories | RRID:IMSR_JAX:016223 | |

*Continued on next page*

*Continued*

| Reagent type (species) or resource | Designation | Source or reference | Identifiers | Additional information |
|---|---|---|---|---|
| Strain, strain background (Sst Cre, C57Bl6/J background) | Sst$^{tm2.1(cre)Zjh}$/J | Jackson Laboratories | RRID:IMSR_JAX:013044 | |
| Strain, strain background (Pvalb Cre, C57Bl6/J background) | Pvalb$^{tm1(cre)Arbr}$/J | Jackson Laboratories | RRID:IMSR_JAX:017320 | |
| Strain, strain background (Cck-IRES Cre, C57Bl6/J background) | Ccktm1.1(cre)Zjh/J | Jackson Laboratories | RRID:IMSR_JAX:012706 | |
| Transfected construct (*M. musculus*) | AAV2-hSyn-DIO-hM4D(Gi)-mCherry | PMID:21364278 | RRID:Addgene_44362 | |
| Antibody | (goat polyclonal) anti-mouse Phox2b antibody | R and D Systems | RRID:AB_10889846 | (1:100 dilution) |
| Antibody | (rat monoclonal) anti-mouse GAD67 antibody, close 1G10.2 | Millipore Sigma | RRID:AB_2278725 | (1:250 dilution) |
| Antibody | (mouse monoclonal) anti-mouse Parvalbumin antibody | Millipore Sigma | RRID:AB_2174013 | (1:250 dilution) |
| Antibody | (rabbit polyclonal) anti-mouse CCK-AR antibody | R and D Systems | RRID:AB_2275486 | (1:250 dilution) |
| Antibody | (mouse monoclonal) anti-Somatostatin antibody | Santa Cruz Biotechnology | RRID:AB_831726 | (1:200 dilution) |
| Antibody | (rabbit polyclonal) anti-lucifer yellow antibody | ThermoFisher | RRID:AB_2536190 | (1:500 dilution) |
| Antibody | (donkey polyclonal) anti-rabbit AlexaFluor 488 | Jackson Immunoresearch | 711-545-152 | (1:500 dilution) |
| Antibody | (donkey polyclonal) anti-mouse AlexaFluor 647 | Jackson Immunoresearch | 715-605-150 | (1:500 dilution) |
| Sequence-based reagent | RNAScope Probe-Gad1 | ACDBio | 400951-C3 | 1:50 |
| Sequence-based reagent | RNAScope Probe-Slc6a5 | ACDBio | 425351-C2 | 1:50 |
| Sequence-based reagent | RNAScope Probe-Slc32a1 | ACDBio | 319191 | 50:1 |
| Commerical assay or kit | RNAscope Fresh Frozen Multiplex Fluorescent Kit | ACDBio | 320851 | |
| Commerical assay or kit | Chromium Single Cell 3' Reagent Kit | 10X Genomics | PN 120237 | Version 2 |
| Chemical Compound, drug | Lucifer Yellow | Sigma | L0259 | 0.2% |
| Chemical compound, drug | Clozapine | Sigma | 1142107 | 1 mg/mL |
| Chemical compound, drug | Strychnine | Sigma | S0532 | 2 µM |
| Chemical compound, drug | Gabazine | Tocris | 1262 | 10 µM |
| Chemical compound, drug | CNQX | Tocris | 0190 | 10 µM |
| Chemical compound, drug | PPADS | Tocris | 0625 | 100 µM |
| Chemical compound, drug | 1,3-dimethyl-8-phenyl-xantine (8-PT) | Sigma | P2278 | 10 µM |
| Software, algorithm | Loupe Browser | 10X Genomics | RRID:SCR_018555 | Version 5.0.0 |
| Software, algorithm | Ponemah | DSI | RRID:SCR_01701 | Version 5.32 |
| Software, algorithm | Spike | Cambridge Electronic Design | RRID:SCR_00903 | Version 5.0 |
| Software, algorithm | Prism 7 | GraphPad | RRID:SCR_002798 | Version 7.03 |
| Software, algorithm | pCLAMP 10 | Molecular Devices | RRID:SCR_011323 | Version 10 |

*Continued on next page*

*Continued*

| Reagent type (species) or resource | Designation | Source or reference | Identifiers | Additional information |
|---|---|---|---|---|
| Software, algorithm | ImageJ | NIH | RRID:SCR_003070 | Version 2.0.0 |
| Software, algorithm | Synaptosoft | Mini Analysis Program | RRID:SCR_002184 | |

## Animals

All procedures were performed in accordance with National Institutes of Health and University of Connecticut Animal Care and Use Guidelines. All animals were housed in a 12:12 light dark cycle with normal chow ad libitum if of weaning age. The single-cell RNA-seq experiments used mixed sex wild-type C57BL6/J animals. The *Slc32a1*[Cre] (JAX # 016962) and TdTomato (Ai14) reporter mice (JAX # 007914) were maintained on 129S1/SvlmJ and C57BL6/J backgrounds, respectively, and only F1 pups were used for electrophysiological experiments. Mouse lines used for chemogenetic experiments including *Sst*[Cre] (JAX # 013044), *Pvalb*[Cre] (JAX # 017320), and *Cck*[Cre] (JAX # 012706) lines were ordered from a congenic C57BL6/J background, directly from Jackson Laboratories. *Phox2b*[Cre] (JAX # 016223) mice were maintained on a C57BL6/J background and used solely for antibody specificity confirmation.

## Single-cell isolation

Animals were euthanized under ketamine/xylazine anesthesia and brainstem slices were prepared using a vibratome in ice cold, high-sucrose slicing solution containing (in mM): 87 NaCl, 75 sucrose, 25 glucose, 25 NaHCO$_3$, 1.25 NaH$_2$PO$_4$, 2.5 KCl, 7.5 MgCl$_2$, 0.5 mM CaCl$_2$, and 5 L-ascorbic acid (equilibrated with 5% CO$_2$-95% O$_2$). Coronal brainstem slices (300 µm thick) were prepared and then immediately enzymatically treated at 34°C with protease XVIII (6 mg/mL, Sigma) for 6 min. After enzyme incubation, slices were washed three times in cold dissociation solution and then transferred to an enzyme inhibitor mix containing trypsin inhibitor (10 mg/mL, Sigma) and bovine serum albumin (BSA, 10 mg/mL, Sigma) in cold sucrose dissociation solution. Next, we isolated the parafacial region which included cells within ~100 µm from the ventral surface and extended ~600 µm medially from the border of the trigeminal nucleus and ~500 µm rostrally from the caudal end of the facial nucleus. Motor neurons, adrenergic C1 cells and raphe neurons were identified based on expression of cell type specific markers and excluded from analysis. These tissue chunks were then warmed to 34°C for 10 min before trituration. A single cell suspension was achieved by trituration using a 25 and 30 gauge needles sequentially, attached to a 2 mL syringe. Samples were triturated for an average of 5 min. Immediately after, the samples were placed back on ice and filtered through a 30-micron filter (Miltenyi Biotech) into sterile microcentrifuge tubes for cell viability assessment.

## Single-cell RNA sequencing

Cell viability for each sample was assessed on a Countess II automated cell counter (ThermoFisher), and 12,000 cells were loaded for capture onto an individual lane of a Chromium Controller (10X Genomics). Single cell capture, barcoding and library preparation were performed using the 10X Chromium platform according to the manufacturer's protocol (#CG00052) using version 2 (V2) chemistry. cDNA and libraries were checked for quality on Agilent 4200 Tapestation, quantified by KAPA qPCR. All libraries were sequenced on individual lanes of an Illumina HiSeq4000 targeting 6000 barcoded cells with an average sequencing depth of 50,000 reads per cell.

## scRNA-seq data processing, quality control, and analysis

Illumina base call files for both libraries were converted to FASTQs using bcl2fastq v2.18.0.12 (Illumina) and FASTQ files were aligned to the mm10 (GRCh38.84, 10X Genomics mm10 reference 2.1.0) using the version 2.2.0 Cell Ranger count pipeline (10X Genomics), resulting in two gene-by-cell digital count matrices. Source code for FASTQ files is available on GitHub at: https://github.com/TheJacksonLaboratory/ventral-parafacial-neuron-scrnaseq or through GEO accession GSE153172. Downstream analysis was performed using Scanpy (v1.4.6) (*Wolf et al., 2018*). Individual libraries were subjected to quality control and filtering independently. Putative doublets were first removed using Scrublet on the raw matrixes (*Wolock et al., 2019*). Then, for each matrix, cells

containing fewer than 800 genes, more than 50 hemoglobin transcripts, or more than 20% mtRNA content were excluded from downstream analyses. Genes present in five or fewer cells, with fewer than 10 total counts were also excluded. For the dataset depicted in this work, we used two lanes of 10X Chromium chip with the following conditions: P10 C57BL6/J pups and 4-OH Tamoxifen treated P10 C57BL6/J pups. We used both of these conditions to create a comprehensive control database to compare Cre dependent and inducible lines with a C57BL6/J background strain. The individual filtered matrices (containing 3345 and 9054 cells, respectively) were concatenated together resulting in an initial aggregated counts matrix of 12,399 cells by 15,923 genes. This aggregated counts matrix was normalized by the total number of counts per cell then multiplied by the median number of counts across all cells, log2 transformed, and lastly scaled to zero mean and unit variance columnwise.

The 2000 most highly variable genes computed using 'scanpy.pp.highly_variable_genes' with 'flavor="cell_ranger'' were selected for the computation of principal components (PCs). Genes related to cell cycle, stress response, the Y-chromosome, hemoglobin as well as ribosomal, mitochondrial, and the gene Xist were excluded from this list of highly variable genes prior to the computation of PCs. Briefly, the list of cell cycle genes was adapted from *Giotti et al., 2019*, using all annotated genes except those with labels 'Other', 'Function known but', and 'Uncharacterized', whereas the stress response genes were adapted from *O'Flanagan et al., 2019*. The first 25 PCs were computed and were used to create a k = 16 batch-balanced nearest-neighbor graph using BBKNN (*Polański et al., 2020*) measured by cosine distance. A 2D UMAP embedding (https://arxiv.org/pdf/1802.03426.pdf) was subsequently generated using this graph ('min_dist = 0.5'). Initial clusters were assigned via the Leiden community detection algorithm (*Traag et al., 2019*) at 0.7 resolution on this k-NN graph, resulting in 20 initial clusters (*Figure 1—figure supplement 1*).

To further analyze the neuronal populations, all cells were first classified as neuronal or non-neuronal using a simple two-state Gaussian mixture model (*Mickelsen et al., 2019*), and neuronal cells were further classified as inhibitory or excitatory using a three-state Gaussian mixture model. Briefly, the median expression of *Snap25*, *Syp*, *Tubb3*, and *Elavl2* in each cluster was used to fit the first model and classify the clusters as neuronal (high expression) and non-neuronal (low expression). Median expression of *Slc17a6* and *Slc32a1* was then used to classify the neurons as *Slc17a6*$^{high}$ (excitatory), *Slc32a1*$^{high}$ (inhibitory), or other. Inhibitory and excitatory neurons were subsequently reanalyzed separately (*Figure 1—figure supplement 1*).

The raw expression matrices of the 210 excitatory and 464 inhibitory neurons were re-normalized, batch corrected, and embedded with UMAP independently as described above, using only 1000 highly variable genes. Clustering with Leiden community detection led to four excitatory and seven inhibitory neuron clusters. In both analyses, a small population of *Slc17a6* high/*Slc32a1* high cells emerged (n = 28); it is unclear whether these cells are doublets, but due to the limited number of such cells, this unique population was discarded from both analyses, resulting in 197 excitatory and 445 inhibitory neurons analyzed. Markers genes for each cluster in each analysis were computed using the Wilcoxon-ranked sum test in a one-versus-rest fashion using the 'scanpy.tl.rank_genes_groups' function.

## Immunohistochemistry

Mice (*Phox2b*$^{Cre}$::TdT, *Slc32a1*$^{Cre}$::TdT, *Slc32a1*$^{Cre}$ infected with virus) were transcardially perfused with 20 mL of room temperature phosphate buffered saline (PBS, pH 7.4) followed by 20 mL of chilled 4% paraformaldehyde (pH 7.4) in 0.1 M phosphate buffered saline. The brainstem was then removed from the animal and post-hoc fixed for 24 hr. After, 150 μm slices were made using a Zeiss VT100S vibratome. In the case of live slices from electrophysiology recordings, individual slices were placed in chilled 4% paraformaldehyde for at least 24 hr before processing. Free floating slices were then incubated in a 0.5% Triton-X/PBS solution for 45 min to permeabilize the tissue. The slices remained in a 0.1% Triton-X/10% Fetal Bovine Serum (FBS, ThermoFisher)/PBS solution for a 12 hr primary antibody incubation of selected primary antibody (see **Key Resource Table**). The tissue was then washed three times in 0.1% Triton-X/10% FBS/PBS solution; the secondary antibody was incubated with the tissue after the third wash for 2 hr (see **Key Resource Table**). The tissue was then washed three times in PBS before mounting on precleaned glass slides with Prolong Diamond with DAPI (ThermoFisher). Imaging of brain slices was achieved with a Leica SP8 confocal microscope.

## Fluorescent in situ hybridization

To prepare fresh frozen slice, two week old $Slc32a1^{Cre}$::TdT mice were anesthetized with isoflurane, decapitated, and brainstem tissues were rapidly frozen with dry ice and embedded with OCT compound. Brainstem slices (14 μm thick) containing the RTN were cryosectioned and collected onto SuperFrost Plus microscope slides. Slices were fixed with 4% paraformaldehyde and dehydrated with ethanol. This tissue was processed with the instruction of RNAscope Multiplex Fluorescent Assay (ACD, 320850); the probes used in our study were designed and validated by ACD. Confocal images were obtained using a Leica Sp8 and confocal image files containing image stacks were loaded into ImageJ (version 2.0.0, NIH, RRID:SCR_003070).

## Acute brainstem slice preparation and in vitro electrophysiology

Slices containing the RTN were prepared as previously described (*Kuo et al., 2016*). In short, mice were anesthetized by administration of ketamine (375 mg/kg, I.P.) and xylazine (25 mg/kg; I.P.) and rapidly decapitated; brainstems were removed and transverse brainstem slices (250–300 μm) were cut using a microslicer (DSK 1500E; Dosaka) in ice-cold substituted Ringer solution containing the following (in mM): 260 sucrose, 3 KCl, 5 $MgCl_2$, 1 $CaCl_2$, 1.25 $NaH_2PO_4$, 26 $NaHCO_3$, 10 glucose, and 1 kynurenic acid. Slices were incubated for 30 min at 37°C and subsequently at room temperature in a normal Ringer's solution containing (in mM): 130 NaCl, 3 KCl, 2 $MgCl_2$, 2 $CaCl_2$, 1.25 $NaH_2PO_4$, 26 $NaHCO_3$, and 10 glucose. Both substituted and normal Ringer's solutions were bubbled with 95% $O_2$ and 5% $CO_2$ (pH = 7.30).

## Electrophysiological recordings

Individual slices containing the ventral parafacial region were transferred to a recording chamber mounted on a fixed-stage microscope (Olympus BX5.1WI) and perfused continuously (~2 ml/min) with a bath solution containing (in mM): 130 NaCl, 3 KCl, 2 $MgCl_2$, 2 $CaCl_2$, 1.25 $NaH_2PO_4$, 26 $NaHCO_3$, and 10 glucose (equilibrated with 5% $CO_2$; pH = 7.3). All recordings were made with an Axopatch 200B patch-clamp amplifier, digitized with a Digidata 1322A A/D converter and recorded using pCLAMP 10.0 software (RRID:SCR_011323). The firing response of chemosensitive RTN neurons to 10% $CO_2$ (duration of ~5 min) was assessed at room temperature (~22°C) in the cell-attached voltage-clamp configuration (seal resistance >1 GΩ) with holding potential matched to resting membrane potential (Vhold = −60 mV) and with no current generated by the amplifier (Iamp = 0 pA). Patch electrodes had a resistance of 5–6 MΩ when coated with Sylgard 184 and filled with a pipette solution containing the following (in mM): 120 $KCH_3SO_3$, 4 NaCl, 1 $MgCl_2$, 0.5 $CaCl_2$, 10 HEPES, 10 EGTA, 3 Mg-ATP and 0.3 GTP-Tris, 0.2% Lucifer yellow (pH 7.30). Firing rate histograms were generated by integrating action potential discharge in 10 to 20 s bins using Spike 5.0 software (RRID:SCR_000903). In a subset of experiments, we obtained whole-cell access to characterize input resistance over voltages ranging from −20 to −80 mV by injecting current steps (1 s) of varying amplitudes (−100 pA to −20 pA). Also, in the whole cell configuration, we filled cell types of interest with Lucifer Yellow for *post hoc* immunohistochemical identification using cell type specific markers.

Spontaneous synaptic currents were characterized in the absence of TTX using a Cs-based pipette solution containing the following (in mM): 135 $CsCH_3SO_3$, 10 HEPES, 1 EGTA, 1 $MgCl_2$, 3.2 TEA-Cl, 5 Na-phosphocreatine, 4 Mg-ATP, and 0.3 Na-GTP (pH 7.3 using CsOH). To record spontaneous IPSCs (sIPSCs), cells were held at the reversal potential for AMPA-mediated excitatory synaptic currents (sEPSCs; Ihold 0 mV) and confirmed with bath application of GABA and glycine blockers. To record EPSCs, cells were held at −60 mV. Although this voltage is positive to the $Cl^-$ reversal potential under our experimental conditions, potential contaminating IPSCs could easily be excluded from analysis based on the direction of synaptic currents; inward for EPSCs and outward for IPSCs. We also pharmacologically confirmed EPSCs as glutamatergic at the end of each experiment by bath application of 6-cyano-7- nitroquinoxaline-2,3-dione (CNQX). Spontaneous EPSCs and IPSCs were analyzed using the Mini Analysis Program (Synaptosoft) and detected events based on amplitude (minimum 5 pA) and characteristic kinetics (fast rising phase followed by a slow decay). Each automatically detected event was also visually inspected to exclude obvious false responses. All whole-cell recordings had an access resistance (Ra) <20 MOhm, recordings were discarded if Ra varied 10% during an experiment, and capacitance and Ra compensation (70%) were used to minimize

voltage errors. A liquid junction potential of −10 mV ($KCH_3SO_3$) or +11 mV ($CsCH_3SO_3$) was corrected off-line.

## RTN viral injections

Adult $Slc32a1^{Cre}$, $Sst^{Cre}$, $Pvalb^{Cre}$, and $Cck^{Cre}$ mice (>20 g) were anesthetized with 3% isoflurane. The right cheek of the animal was shaved and an incision was made to expose the right marginal mandibular branch of the facial nerve. The animals were then placed in a stereotaxic frame and a bipolar stimulating electrode was placed directly adjacent to the nerve. Animals were maintained on 1.5% isoflurane for the remainder of the surgery. An incision was made to expose the skull and two 1.5 mm holes were drilled left and right of the posterior fontanelle, caudal of the lambdoidal suture. The facial nerve was stimulated using a bipolar stimulating electrode to evoke antidromic field potentials within the facial motor nucleus. In this way, the facial nucleus on the right side of the animal was mapped in the X, Y, and Z direction using a quartz recording electrode. The viral vector, AAV2-hSyn-DIO-hM4D(Gi)-mCherry (Addgene #44361, titer 5.3 × $10^{12}$ GC/mL), was loaded into a 1.2 mm internal diameter borosilicate glass pipette on a Nanoject III system (Drummond Scientific). One 10 µL injection of virus per side was delivered at least −0.02 mm ventral to the Z coordinates of the facial nucleus, to ensure injection into the RTN. These same coordinates were used for the left side of the animal. In all mice, incisions were closed with nylon sutures and surgical cyanoacrylate adhesive. Mice were placed on a heated pad until consciousness was regained. Meloxicam (1.5 mg/kg) was administered 24 and 48 hr postoperatively. Plethysmography was performed 2 weeks after viral injection. The location of all injection sites were later confirmed by post hoc histological analysis (*Figure 5—figure supplement 1*).

## Unrestrained whole-body plethysmography

Respiratory activity was measured using a whole-body plethysmograph system (Data Scientific International; DSI), utilizing a small animal chamber maintained at room temperature and ventilated with room air or carbogen mixtures at a constant flow rate of 1.16 L/min. Following recovery from surgery, $Slc32a1^{Cre}$, $Sst^{Cre}$, $Pvalb^{Cre}$, and $Cck^{Cre}$ mice injected with AAV were individually placed into a chamber and allowed 1 hr to acclimate prior to the start of an experiment. Respiratory activity was recorded using Ponemah 5.32 software (DSI) for a period of 20 min in room air immediately after I.P. injection of saline or clozapine followed by exposures to 0, 3, 5, and 7% $CO_2$ (balance $O_2$) (10 min/condition). For the chemogenetic experiments, we sequentially tested saline followed by clozapine on the ventilatory response to $CO_2$ in the same animals on the same day (~2.5 hr between trials). Plethysmography experiments began immediately after injection of either saline or clozapine. Parameters of interests including respiratory frequency (FR, breaths per minute), tidal volume (VT, measured in mL; normalized to body weight and corrected to account for chamber and animal temperature, humidity, and atmospheric pressure), and minute ventilation (VE, mL/min/g) were measured during a 20 s period of relative quiescence, confirmed with synchronous video monitoring, after 5 min of exposure to each condition. All experiments were performed between 9 a.m. and 6 p.m. to minimize potential circadian effects.

## Statistics

Data are reported as mean ± SE. Power analysis was used to determine sample size, all data sets were tested for normality using Shapiro-Wilk test, and outlier data points were identified by the Grubbs test and excluded from analysis. Statistical comparisons were made using t-test, Wilcoxon-ranked sum test, or one-way or two-way simple or repeated measures ANOVA or ANCOVA followed by Tukey's multiple comparison tests as appropriate. The specific test used for each comparison is reported in the figure legend and all relevant values used for statistical analysis are included in the results section.

## Acknowledgements

This work was supported by funds from the National Institutes of Health Grants HL104101 (DKM), HL137094 (DKM), NS099887 (DKM), F31HL142227 (CMC) and F31NS120467 (BMM).

## Additional information

### Funding

| Funder | Grant reference number | Author |
|---|---|---|
| National Institutes of Health | HL104101 | Daniel K Mulkey |
| National Institutes of Health | HL137094 | Daniel K Mulkey |
| National Institutes of Health | NS099887 | Daniel K Mulkey |
| National Institutes of Health | HL142227 | Colin M Cleary |
| National Institutes of Health | F31NS120467 | Brenda M Milla |

The funders had no role in study design, data collection and interpretation, or the decision to submit the work for publication.

### Author contributions

Colin M Cleary, Data curation, Formal analysis, Funding acquisition, Writing - review and editing; Brenda M Milla, Data curation, Funding acquisition, Investigation, Writing - review and editing; Fu-Shan Kuo, Conceptualization, Data curation, Formal analysis, Writing - review and editing; Shaun James, William F Flynn, Paul Robson, Data curation, Formal analysis, Writing - review and editing; Daniel K Mulkey, Conceptualization, Supervision, Funding acquisition, Writing - original draft, Project administration, Writing - review and editing

### Author ORCIDs

Colin M Cleary  https://orcid.org/0000-0003-0305-1324
William F Flynn  https://orcid.org/0000-0001-6533-0340
Paul Robson  https://orcid.org/0000-0002-0191-3958
Daniel K Mulkey  https://orcid.org/0000-0002-7040-3927

### Ethics

Animal experimentation: All procedures were performed in accordance with National Institutes of Health and University of Connecticut Animal Care and Use Guidelines (protocols A19-048 and A20-016).

### Decision letter and Author response

Decision letter https://doi.org/10.7554/eLife.60317.sa1
Author response https://doi.org/10.7554/eLife.60317.sa2

## Additional files

### Supplementary files

• Transparent reporting form

### Data availability

Raw and processed scRNA-seq data are available through the Gene Expression Omnibus (accession GSE153172) and analysis code is available on GitHub. Analysis of FISH, electrophysiology, and respiratory activity data was done using standard software and no custom code was written.

The following dataset was generated:

| Author(s) | Year | Dataset title | Dataset URL | Database and Identifier |
|---|---|---|---|---|
| Cleary CM, Kuo FS, James S, Flynn WF, Robson P, Mulkey DK | 2020 | Ventral parafacial inhibitory neurons regulate baseline breathing by a mechanism involving disinhibition | https://www.ncbi.nlm.nih.gov/geo/query/acc.cgi?acc=GSE153172 | NCBI Gene Expression Omnibus, GSE153172 |

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
