## [Decision Letter]

**Acceptance summary:**

A parafacial region of the medulla called the retrotrapezoid nucleus (RTN) is an important respiratory control center that responds vigorously to CO_2_ changes, contributes to arterial PCO_2_ homeostasis and regulates breathing during sleep. Within the RTN, glutamatergic neurons function as respiratory chemoreceptors by regulating breathing in response to changes in tissue CO_2_/H^+^. However, only very limited attention has been devoted so far to investigate the role of inhibitory neurons in RTN function. The present study by Mulkey's lab explores the possible contribution of local inhibitory neurons to the activity and acid-sensitivity of RTN neurons. The authors used multiple technologies (RNAscope, RNA seq, electrophysiology, chemogenetics) in vivo and in vitro to specifically target inhibitory interneurons located at the anatomical site of the RTN. They show that this region contains a limited diversity of inhibitory neurons. Among these, only Sst-expressing inhibitory interneurons appear inhibited by CO_2_/H^+^ and to send inhibitory inputs on the RTN chemosensitive neurons. By doing so these neurons finely tune resting respiratory drive according to CO_2_ changes in low physiological ranges. These new findings establish parafacial SST-expressing inhibitory neurons as important regulators of baseline breathing and contributor of RTN chemoception. Furthermore, this paper brings valuable mechanistic insights into how loss of inhibition within the RTN might disrupt breathing in pathological conditions such as in the Dravet syndrome.

**Decision letter after peer review:**

Thank you for submitting your article "Ventral parafacial inhibitory neurons regulate baseline breathing by a mechanism involving disinhibition" for consideration by *eLife*. Your article has been reviewed by 3 peer reviewers, and the evaluation has been overseen by a Reviewing Editor and Ronald Calabrese as the Senior Editor. The following individuals involved in review of your submission have agreed to reveal their identity: Clément Menuet (Reviewer #1); Natasha N Kumar (Reviewer #2); Patrice Guyenet (Reviewer #3).

The reviewers have discussed the reviews with one another and the Reviewing Editor has drafted this decision to help you prepare a revised submission.

Essential revisions:

More specifically, I would like to draw your attention to the fact that the three reviewers have carefully examined your manuscript and while they found a great interest for your results they also pointed out on several important aspects of the paper that require modifications. Their reviews are detailed below but I would like to emphasize on two major concerns that must be adequately addressed before suitability for publication in *eLife* could be considered:

First, they all noted a lack of precision in the description of the different methods used, in the size of the samples analyzed and illustrated in the figures, the parameters measured, etc… In addition, the experimental conditions in some cases are not always clear. More importantly, the authors must provide evidences for the specificity and selectivity of the transgenic lines used and the Dreadd and TdTomato expression and viral injections. In general, the methods need to be more precise and better explained throughout the text.

Second, the transcriptomic part, as it stands, does not really add much to the physiological question and remains a little apart from the rest of the study. For instance, it does not help in the identification of the neuronal cluster involved in the acid sensitivity of the RTN, and potential specific genes that could be indicative of a specific population of inhibitory neurons playing a role here are not used in the subsequent experiments presented after. Thus, either you put less emphasis in this part of the results (and even consider presenting these data at the end of the result section) or you provide a deeper analysis and a better link with the other presented data that would definitively bring a level of specificity and understanding that would greatly enhance the quality of the work and the interest of providing such genetic findings.

Note that acceptability will have to be reassessed upon submission of the revised version.

*Reviewer #1:*

The proposed study is a Research Advance paper aimed at investigating the role of inhibitory parafacial neurons in the control of breathing. The authors first performed single cell RNA sequencing of the RTN/parafacial region to identify clusters of excitatory and inhibitory neurons. Then they showed using whole-cell recordings in slices that about 25% of recorded inhibitory parafacial neurons are tonically active and inhibited by hypercapnia, and that chemosensitive RTN neurons receive tonic inhibitory inputs that are decreased by hypercapnia, and excitatory inputs that are increased by hypercapnia. Chemogenetic inhibition of inhibitory parafacial neurons increased normocapnic/normoxic and normocapnic/hyperoxic breathing but not hypercapnic breathing in mice in vivo.

To my knowledge, this is the first study investigating the specific role of inhibitory parafacial neurons in the control of breathing activity. The main finding, that inhibitory parafacial neurons enable fine tuning of resting respiratory drive according to variations of CO2 in the low physiological range, is very interesting to the field, and potentially to a broader audience. The study is well-designed, using state-of-the-art techniques, and the paper is well written.

1. The Results section starts with single cell RNA sequencing data of RTN/parafacial neurons, identifying clusters of glutamatergic and GABA/glycinergic neurons. In itself, this is very valuable data, which could justify a stand-alone paper with further analysis. However, starting the Results section with these data is a bit awkward, as they are poorly linked to the rest of the study. Indeed, the authors provide new subclasses of inhibitory neurons, but then focus the rest of the study on overall inhibitory neurons without attempting to link their functional results with the subclasses of neurons molecularly identified. The transition line 164, to show that TdT is expressed in overall inhibitory neurons, without using tools for further specificity as one would expect after reading the previous paragraph on scRNAseq, is somewhat frustrating.

The paragraph starting line 172 shows that a subset of parafacial inhibitory neurons are CO2/H^+^ sensitive, and finishes with a statement (lines 188-190) that the identity of this subset of neurons remains to be determined. Testing whether this subset corresponds to a specific cluster of inhibitory neurons identified in the scRNAseq data presented just before would seem the obvious next step for the present study. The authors could have done immunohistochemistry on the neurons patched, as several of the clusters identified have markers for which antibodies exist, or better (but harder) they could have aspirated the patched cytoplasm and performed qPCR to reach the same results. Such data would not only provide a natural link between the first two experimental sets performed, but also provide more detailed information on the inhibitory parafacial neurons that are CO2/H^+^ sensitive that would significantly enhance the impact of this study.

With these comments in mind, if no further experiment is done for this paper, it might be better to move the scRNAseq data to the end of the Results section, to open the work towards more specificity in future studies. On that line, the end of the sentence lines 168-170, which closes the scRNAseq data presentation, is more suited for the end of the Results section, rather than the beginning.

2. In a previous article by the same group (Kuo FS et al., *eLife* 2019, DOI: https://doi.org/10.7554/*eLife*.43387), mice expressing a Dravet syndrome-associated Scn1a missense mutation conditionally in inhibitory neurons presented breathing alterations including diminished chemosensitivity, with hypo-excitable inhibitory neurons and hyper-excitable excitatory neurons in the RTN. The present study is proposed as Research Advance of this previous article. To strengthen the continuity with the previous study, the authors could have analysed the expression of Snc1a in their scRNAseq data, to identify whether Scn1a is expressed in a cluster of inhibitory neurons, and link it with the cluster that is responsive to CO2/H^+^. This way, the link would be more evident, and the present study would really make use of the scRNAseq data. In the current state of this study, I somewhat fail to see its strong and direct continuity with the previous work justifying a Research Advance format, it would maybe be better suited as a traditional stand-alone Research Article.

3. Was synaptic isolation performed when testing the CO2/H^+^ sensitivity of parafacial inhibitory neurons? Did you also block the usual gliotransmitters? The methods section is not clear, please provide more details, as whether these neurons are intrinsically sensitive to CO2/H^+^ or not is critical for interpretation of these data.

*Reviewer #2:*

General assessment: Previous work from the Dan Mulkey group demonstrated that mice expressing a Scn1a mutation exhibit a Dravet syndrome associated phenotype, wherein inhibitory neurons of the RTN were less excitable, whereas glutamatergic chemosensitive neurons were hyperexcitable. In this manuscript, Cleary and colleagues used multiple technologies (RNAscope, RNA seq, electrophysiology, chemogenetics) to target inhibitory interneurons located at the anatomical site of the retrotrapezoid nucleus, which harbours glutamatergic respiratory chemosensory neurons. It is now established that inputs from other chemoreceptor neurons as well as astrocytes-derived paracrine signals, contributes to RTN chemoreceptor drive. Here they show that RTN neurons receive tonic inhibitory inputs from non-chemosensing VGAT neurons in the parafacial region. The authors provide careful control conditions. The manuscript figures are presented well experimental procedures and results are sound and support the conclusions. This work warrants publication in *eLife* in principle, the results are novel both technically and from a biological perspective, upon the condition that the authors adequately address the reviewers questions.

1. In the first paragraph of the discussion, the authors conclude that inhibitory inputs to chemosensing RTN neurons are withdrawn/inhibited during exposure to high CO2. The net result is a lessening of GABA restraint on chemosensing neurons and an increase in their neural drive and neurotransmitter release. Another valid interpretation could be that increased neurotransmission from other tonic inputs to the RTN (arising from chemoreceptors activated during high CO2 exposures) contributes to the ventilatory response? Ie the VGAT inhibitory input isn't withdrawn but rather contributes much less to RTN drive during exposure to high CO2. Please include this interpretation, and other alternatives in the discussion.

2. DREADD-mediated inhibition of inhibitory RTN neurons increased baseline minute ventilation but had no effect on ventilatory responses to CO2. Figure 6D-E – most columns do not include data from n=11 mice. Please explain why certain mice/replicates were excluded from the dataset. Figure 6 figure legend is incomplete – there is no description of the data shown for control Vgatcre (Figure 6F-H). Also, there are two Figure 6F.

3. CO2/H^+^ synaptic properties of RTN neurons studies (page 8, line 193):

"As previously defined (26), RTN neurons were considered chemosensitive if they show some level of spontaneous activity under control conditions and a robust firing rate response to 10% CO2 (1.6 {plus minus} 0.36 Hz; N=10)." – 10% CO2 FR response of 1.6 {plus minus} 0.36 Hz does not seem 'robust' when compared to other previous studies. For example, Wang S et al. 2013 Figure 2D (citation #64), demonstrates FR at pH 7 (~8% CO2) to be >4 Hz for both TypeI and TypeII dissociated RTN neurons. Wouldn't you expect FR to be >>4Hz in 10% CO2, for RTN neurons in an intact brain slice?

4. Figure 2 and Figure 2 legend:

Figure 2B – error bars appear to be missing for this n=4 quantified data.

Figure 2Aii is an enlarged version of the image at bregma -6.05mm. You could omit this, I don't think it adds anything to the paper. Alternatively, magnify further, use arrows to draw attention to cells of interest.

Figure 2 legend: include abbreviations (7N, b)

5. Figure 3C legend: It is not clear what the summary data corresponds to. Indicate this in the figure legend as follows: ' firing responses of CO2/H^+^ -inhibited (n=20, blue), -insensitive (n=53, grey) and -activated (n=7, red). What parameters do the authors use, to define inhibited versus activated neurons?

6. Figure 5 – please confirm that there are no outliers in these values. One replicate seems to have a much higher baseline sEPSC frequency than the others.

7. Supplementary Figure 1 figure legend states that the colors of distributions correspond to classes shown in Figure 1A however Figure 1A displays only 3 colours, whereas Supp Figure 1D displays 5 colours/classes. Please clarify.

8. Authors used RNA seq to identify differentially expressed and coregulated genes in RTN neurons in order to infer biological meaning for further studies on this preeminent central chemoreceptor population. Single cell isolation method and RNA-seq methodology was extremely detailed, valuable information regarding inclusion and exclusion criteria were included. The authors use tSNE to investigate segmentation, clustering of genes in the pFRG transcriptome. The data are convincing, since it is well established using other methods, that ~50% of glutamatergic RTN neurons express galanin. Could the authors clarify how they minimised batch effect, which can occur during the experiment, the RNA library preparation, or the sequencing run?

There are 20 global clusters (from 2000 genes) – the distribution of cells/cluster and genes/cluster is illustrated in Supp Figure 1C. The basis for the clustering was not clear – could the authors clarify this and make it clear in the text. It appears (from Supp Figure 1 B) that each cluster is defined by related genes that display a high level of expression within the cluster.

*Reviewer #3:*

RTN is a small cluster of lower brainstem neurons with a well-described molecular signature. These neurons respond vigorously to CO2 in vivo, contribute to arterial PCO2 homeostasis and maintain breathing automaticity during sleep. Their synaptic inputs are not well identified however and the way in which they encode brain PCO2 in vivo is complex and much debated. Several mechanisms have been proposed (intrinsic pH sensitivity of RTN neurons, astrocyte-mediated paracrine effect of acid, paradoxical acid-mediated contraction of microvessels and synaptic inputs from acid-activated neurons, e.g. serotonergic). The present study explores the possible contribution of local inhibitory neurons to the activity and acid-sensitivity of RTN neurons. Consistent with prior EM evidence, the results show that RTN neurons receive GABA/glycinergic inputs. The contribution of a subset of this input to the CO2 response of RTN neurons is suggested by the results of experiments in slices but the existence of this mechanism is not convincingly supported by the in vivo experiments.

The authors confirm the transcriptome of RTN neurons and describe that of several subsets of inhibitory neurons (likely GABA- and glycine-ergic) that reside in their vicinity. This work greatly extends existing observations regarding these local inhibitory neurons. Unfortunately this work did not help identify the particular inhibitory interneurons postulated to be CO2-responsive.

Next, the authors show that a subset of these inhibitory neurons are mildly inhibited by acidification in slices. The effects are small and variable from cell to cell but suggest that CO2-triggered disinhibition could perhaps contribute to the overall excitatory effect of CO2 on RTN neurons.

This hypothesis is then tested, by determining whether chemogenetic inhibition of these inhibitory interneurons enhances breathing and the respiratory chemoreflex in conscious mice. The authors show is that inhibition of the GABA/glycine interneurons (via a Gi-coupled DREAAD) in unanesthetized mice elevates breathing slightly at rest but seems to have no effect on the breathing stimulation elicited by CO2. The increase in resting breathing is consistent with prior evidence that breathing is increased when a GABA-receptor antagonist is microinjected into the region containing the RTN neurons (Nattie et al) but this fact does not demonstrate that the increase in breathing is caused by activation of RTN neurons. As the authors acknowledge in the discussion the increase in breathing produced by inhibiting GABA interneurons in this brain region could also result from arousal. The increase in breathing could also result from the disinhibition of neurons other than RTN, for example the C1 neurons.

Finally, the main interpretative difficulty is that inhibition of the GABA/glycine interneurons does not change the respiratory chemoreflex; this is difficult to reconcile with the results obtained in slices and makes one wonder whether the acid inhibition of the GABA interneurons observed in slices is an artifact or at least whether it is a phenomenon that has any relevance to the CO2 response of RTN neurons in vivo.

1. Re: results line 177. Although the authors are probably correct, they need to validate the assumption that their reporter line (Vgatcre::TdT) does indeed "allow for selective targeting of inhibitory neurons in the region of interest". If this has been demonstrated before please give the specific quote. If not, the authors must demonstrate that the vast majority of the tomato-red positive neurons in their region of interest express VGat (transcript or protein) at the relevant postnatal age.

2. Re: lines 240-241. "Specifically, we bilaterally injected AAV2-hSyn-DIO-hM4D(Gi)-mCherry (10 nL/side, Addgene) into the medial portion of the RTN in VGAT-cre mice". The authors need to also demonstrate here that VGat+-neurons were selectively transduced (i.e. that mCherry and VGat were colocalized). The selectivity of this type of approach is typically overrated (ectopic expression of Cre and Cre-independent effects of the DIO viral prep) and must be verified.

3. Re: results lines 177-183. The defining criteria for pH sensitivity is fine but the results are less than impressive: 20% of neurons were inhibited to some degree (-20% or more) when exposed to a fairly severe acidification (pH 7.0), 66% were unaffected and 9 % were judged to be excited. Figure 3C should be replaced with a distribution histogram summarizing the % response of all the VGAt neurons sampled (e.g. 10% effect binning). Small positive or negative effects of acidification on unit-activity in slices have frequently been reported in other brainstem areas such as the nucleus of the solitary tract. Cherry-picking neurons that are excited and those that are inhibited is not terribly convincing unless this grouping correlates with an identifiable biochemical or structural marker, which was not the case here.

4. Re: lines 248-249 and discussion. The increase breathing could have many causes and may not have nothing to do with CO2 chemoreception. For example, selective activation of the rostral C1 neurons produces a very strong increase in breathing in unanesthetized mice and these neurons are tonically inhibited by GABA-ergic input (the basis of the baroreflex). The increased breathing noted at baseline (no CO2 added) could also denote arousal from sleep, an effect produced by activation of the C1 neurons, the RTN itself and perhaps other neurons in the area.

[Editors' note: further revisions were suggested prior to acceptance, as described below.]

Thank you for resubmitting your work entitled "Somatostatin-expressing parafacial neurons regulate baseline breathing" for further consideration by *eLife*. Your revised article has been reviewed by 3 peer reviewers, and the evaluation has been overseen by a Reviewing Editor and Ronald Calabrese as the Senior Editor.

This is an unusual decision letter. The situation is as follows. Two of the original reviewers declined to review the manuscript. We decided to solicit extra reviewers. The new review team judge that you have successfully addressed all of the concerns from the last round of review. We are therefore happy to publish the manuscript in its current form.

However, the two new reviewers have also raised some new points that they think will improve the manuscript. You therefore have the *option* to address these new comments if you think it would improve the manuscript. I emphasise this is entirely your decision. In accordance with *eLife* policy, because you have addressed the previous concerns, we are happy to publish the manuscript as it stands.

We apologize for the delay in providing this review. But two out of the three initial reviewers declined to re-review the paper and we had to find two other ones. Consequently, you will find some new concerns. However, be aware that your responses to previous comments have been judged adequate and that the additional experiments now included definitively improve the paper. We are happy to publish the paper in this form.

However, please take the time to read carefully all comments and address those that you choose to address. Basically, they highlight the following potential improvements: A better definition of your hypothesis; clarifying the functional connections between the different sets of data; stating unambiguously whether you consider the neurons of interest as part of the RTN/pFRG or not; providing a better description of the inhibitory neurons investigated in terms of distribution and number; and paying attention to the scales in different figures that might be wrong.

*Reviewer #2:*

The article is much improved and I think the authors have responded adequately to my comments and the comments of the other reviewer.

A comprehensive body of work investigating the neurochemical diversity of parafacial inhibitory neurons. In particular the large majority of scl32a1 neurons (GABA of Glycinergic) 37% are chemosensing and in fact exhibit reduced FR in response to extracellular acidosis, 56%were pH insensitive, 7% showed an excitatory response. The acid-inhibited neurons are intrinsically pH sensitive. 5/5 CO2 inhibited cells are SST immunoreactive.

Chemogenetic suppression of Sst+ parafacial neurons increased respiratory frequency and minute ventilation (F1,5=81.06, p = 0.0003) under baseline conditions

Having seen the original manuscript, the addition of the 2 sets of experiments linking the transcriptomic analysis with the electrophysiological in vitro respiratory experiments, and toning down the emphasis on the transcriptomic analysis, make the narrative of the paper clearer.

I agree with Reviewer 1, the transcriptomic analysis comes across as an add on to the data. There is some information that is not relevant to main narrative of this paper however, this additional information is likely of great interest to a broader audience of neuroscientists.

Whilst important to point out the additional experiments required to definitively demonstrate that the inhibitory RNAseq cluster corresponds with the H^+^ inhibited inhibitory neurons in the RTN, Reviewer 1 acknowledges the difficult nature of these experiments. The authors could include this as future directions in the Discussion section.

The Dan Mulkey group adequately addressed both reviewers suggestion to strengthen continuity with the Kuo et al. 2019 study, in order to justify this manuscript as a Research Advance of the previous study. The Kuo et al. paper is pivotal in that it links impairment of inhibitory parafacial neurons with a clinical syndrome.

The response to Reviewer 1 Major Concern 2 is good in that it reveals Scn1a expression is not restricted to inhibitory neurons.

The response to Major Concern 3 is sound; the knowledge that these inhibitory parafacial neurons are intrinsically chemosensitive (similar to phox2b+ parafacial neurons) is of great value to other researchers in the field.

I am satisfied with the authors responses to Reviewer 2 (my comments).

I am satisfied with the authors responses to Reviewers3

I'd like to commend the authors on their production of this impressive body of work

*Reviewer #4:*

It is a well-performed and very interesting study showing that the ventral medullary region of mice, which includes the RTN, contains a diversity of inhibitory neurons (SSt, CCK, and Pvalb). Some of the SSt-expressing neurons were inhibited by hypercapnia/acidosis and chemogenetic inhibition of SSt neurons, including the SSt neurons insensitive and activated by hypercapnia/acidosis, increases the baseline breathing. Besides, the RTN neurons receive inhibitory synaptic transmission at low frequency, which was reduced in response to hypercapnia/acidosis. Therefore, the authors conclude that the ventral medullary SSt neurons contribute to RTN chemoreception and respiratory activity. The topic of this manuscript is original, relevant, and worth studying. Several omissions cloud the interpretation, and I have several concerns that preclude the straightforward interpretation of the data, which have not been adequately considered in the manuscript. The main issue is in the interpretation, or overinterpretation, of the data.

1. It would be essential to define the hypothesis of the present study clearly. The authors "considered" the possibility that ventral medullary neurons in the region of the RTN sense changes in CO2/H^+^ and regulate RTN chemoreception by a mechanism involving disinhibition. However, the authors provided evidence for the ventral medullary inhibitory neurons' involvement in neither the RTN chemoreception nor RTN disinhibition. The authors then propose a novel mode of chemotransduction involving regulation of basal breathing by CO2/H^+^-dependent disinhibition. In fact, the experiments involving the chemogenetic inhibition of SSt neurons did not demonstrate that the RTN neuronal function was affected. The neurons in the ventral medullary region close to RTN project to the pre-Bötzinger Complex (PMID: 26855425), and these projections could be involved in the increases of baseline breathing following SSt neurons chemogenetic inhibition.

2. There are no clear functional connections between some of the results described. The authors described the presence of a diversity of inhibitory neurons in the ventral medullary region and that the inhibition of SSt neurons increased the baseline breathing without affecting the ventilatory responses to hypercapnia/acidosis. On the other hand, there are data showing that RTN neurons receive synaptic inhibition, which was reduced in response to hypercapnia/acidosis. Therefore, the authors need to demonstrate that: i) the three different populations (activated, insensitive, and inhibited) of SSt neurons are projecting to RTN chemosensitive neurons, and: ii) the chemogenetic inhibition of SSt neurons increases the RTN neurons firing frequency by reducing the inhibitory synaptic transmission. The SSt neurons recorded in vitro were labeled, and the projections to the RTN neurons can be revealed in the same slice. Besides, the effects of SSt neurons inhibition (chemogenetics) on the RTN firing frequency and inhibitory synaptic transmission can also be analyzed in in vitro experiments using the transfected animals. It is important to note that without new experiments to prove the role of the inhibitory ventral medullary SSt neurons in the RTN chemosensitivity, the authors should consider presenting the data in two (I – somatostatin-expressing RTN neurons regulate baseline breathing; II – the effects of hypercapnia/acidosis on the synaptic transmission to RTN neurons) different manuscripts.

3. The authors demonstrated that the inhibitory neurons are located close to the RTN chemoreceptors. In fact, it is possible to observe in the provided images that the Tdt cells of Slc32a1 animals are in the RTN. The anatomical nomenclature is highly confusing in the manuscript. There is no evidence that the described inhibitory neurons are outside of the RTN. Parafacial or even pFRG terminologies only add to the confusion regarding the cell types present in this region of the brain and their role. Therefore, the authors studied the inhibitory neurons of the RTN of mice.

4. Is the CO2/H^+^ sensitivity of RTN neurons different before and after the inhibitory synaptic transmission blockade? In other words, what is the contribution of the synaptic disinhibition to the RTN neuronal firing response to hypercapnia/acidosis? The RTN neurons receive inhibitory synaptic inputs with very low frequency (0.2 Hz) and amplitude (12 pA), and hypercapnia/acidosis reduces it by nearly half. Therefore, in which extension such rare and small events contribute to the increases in RTN neurons firing frequency and breathing after SSt neurons chemogenetic inhibition?

5. The authors should provide the number and the rostro-caudal distributions of transfected SSt, CCK and Pvalb neurons, not only the "center" of bilateral virus injections or percentage of neurons, in the chemogenetic experiments. It is an important issue to report considering data reproducibility by other groups, and the extension of the transfected cells in the ventral medulla can provide evidence for different clusters of inhibitory neurons in mice controlling different functions.

6. What about the anatomical distribution of insensitive, activated, and inhibited SSt neurons in the RTN? The authors could provide this information in Figure 2. Are the capacitance values different between these neurons? In this regard, the neurons seem bigger (~100 µm) than other neurons described in the RTN based on the scales provided in Figure 2, panels Aii and Bii. Are the scales right?

7. Please remove the word "tonic" during the description/discussion of RTN neurons' synaptic inhibition. The authors did not evaluate whether or not the inhibitory synaptic events are tonic because there are no respiratory oscillations in the applied in vitro preparation.

8. Is the scale for measured current of the Figure 4 panel A right? The grouped data in panel B show that the amplitude of sEPSCs is -10 pA, but the representative traces show that it is the level of the electrical noise. Besides, the VC traces in Figure 3 panel A are not representing the grouped data of sIPSCs amplitude in panel B. The events are ~ 50 pA, while the grouped data are ~12.5 pA.

9. Please check the Cl^-^ reversal potential; it is unlikely to be – 60 mV using the described intracellular and extracellular solutions.

10. – Discussion section, line 330: Expiratory neurons in the lateral parafacial region do not express Phox2b in rats (PMID: 28004411). Besides, what evidence shows that the expiratory neurons in the lateral parafacial region are not sensitive to hypercapnia/acidosis?

11. Discussion section, line 413: There is no evidence that disinhibition of Phox2b neurons in the lateral parafacial region evokes forced expiration in rats. In fact, previous studies (references: 19 and 53 and PMID: 28004411) demonstrated that bilateral disinhibition of neurons of the lateral parafacial region, which are located more laterally to the Phox2b positive RTN neurons, induces forced expiration in rats.

*Reviewer #5:*

The most important finding of this study is that Sst-expressing neurons in the parafacial region are involved in regulation of baseline respiratory activity via inhibitory effects on the RTN chemoreception. Authors provided evidences that supported this hypothesis.

This manuscript is revised version that has been reviewed by three referees. Authors' responses to comments seemed to be adequate.

First, they showed that the parafacial region contained a limited diversity of inhibitory neurons including somatostatin (Sst)-, parvalbumin (Pvalb)- and cholecystokinin (Cck)-expressing neurons. They showed that a subset of inhibitory parafacial neurons were inhibited by 10% CO2 and the response was retained when purinergic signaling and fast neurotransmission was blocked, suggesting these cells were intrinsically CO2/H^+^ sensitive. Next, they showed that CO2/H^+^ -inhibited cells were Sst-immunoreactive and were not immunoreactive for Pvalb or Cck, whereas CO2/H^+^ -insensitive cells did not express Sst. They also showed that sIPSCs in the RTN chemoreceptor neurons were depressed by 10% CO2. These results suggest that Sst-expressing neurons of inhibitory parafacial neurons were inhibited by CO2/H^+^ and then contributed to hypercapnic response of the RTN chemoreceptor. This hypothesis was further supported by results from in vivo experiment in which chemogenetic suppression of Sst+ parafacial neurons increases baseline breathing.

Although I have no major concern about this paper, please see below for my comments.

1. Page 9, line 212-214, "We found that 5 of 5 CO2/H^+^ -inhibited cells were Sst-immunoreactive (Figure 2Aii) and were not immunoreactive for Pvalb or Cck (data not shown), whereas 0 of 5 CO2/H^+^ -insensitive cells expressed Sst (Figure 2Bii).": This part is most important parts because it was the results indicating that Sst neurons were indeed inhibited by hypercapnic stimulation. Before this part, they showed 48 of 130 (37%) cells that were thought to be inhibitory neurons were inhibited hypercapnic stimulation (page 8). To further clarify properties of these cells, authors investigated 5 cells with combination of whole-cell recordings and immunoreactive examination and showed that cells that were inhibited by hypercapnic stimulation were Sst-immunoreactive. However, this result did not simply mean that all of above 37% neurons were Sst-immunoreactive. Although in vivo experiments supported the idea, is this number (n=5) really enough for the verification? In addition, I suppose that examination for Pvalb or Cck immunoreactivity was performed in cells different from the Sst examination group. Please describe the number of cells tested for Pvalb and Cck. Relating this issue, it would be important to show an example of detailed distribution of Sst (and maybe Pvalb and Cck) immunoreactive cells in the parafacial region together with Phox2b-expressing cells.

2. Page 8, the response to 10% CO2 was retained when purinergic signaling and fast neurotransmission was blocked: I suppose that these experiments were performed separately but not in the presence of purinergic signaling blockers plus fast neurotransmission blockers. I concern that an uncertainty may remain to conclude that parafacial inhibitory neurons are intrinsically CO2/H^+^ -sensitive.

3. Page 10, frequency of sIPSCs: "0.22Hz" seems to be rather low. How such low frequency could affect activity of the RTN chemoreceptor neurons?

---

## [Author Response]

Essential revisions:I would like to draw your attention to the fact that the three reviewers have carefully examined your manuscript and while they found a great interest for your results they also pointed out on several important aspects of the paper that require modifications. Their reviews are detailed below but I would like to emphasize on two major concerns that must be adequately addressed before suitability for publication in eLife could be considered:First, they all noted a lack of precision in the description of the different methods used, in the size of the samples analyzed and illustrated in the figures, the parameters measured, etc… In addition, the experimental conditions in some cases are not always clear. More importantly, the authors must provide evidences for the specificity and selectivity of the transgenic lines used and the Dreadd and TdTomato expression and viral injections. In general, the methods need to be more precise and better explained throughout the text.Second, the transcriptomic part, as it stands, does not really add much to the physiological question and remains a little apart from the rest of the study. For instance, it does not help in the identification of the neuronal cluster involved in the acid sensitivity of the RTN, and potential specific genes that could be indicative of a specific population of inhibitory neurons playing a role here are not used in the subsequent experiments presented after. Thus, either you put less emphasis in this part of the results (and even consider presenting these data at the end of the result section) or you provide a deeper analysis and a better link with the other presented data that would definitively bring a level of specificity and understanding that would greatly enhance the quality of the work and the interest of providing such genetic findings.

We thank the editor and reviewers for their time and thoughtful suggestions. To address the first concerns, we have significantly updated our descriptions and explanations in the results and methods sections. We have also confirmed specificity and selectivity of materials used and those new results are shown in Figure 1—figure supplement 2, Figure 2—figure supplement 1, and Figure 5—figure supplement 1.

To address the second concerns, we performed two additional sets of experiments to identify CO2/H^+^-sensitive inhibitory neurons in vitro and their contribution to respiratory activity in vivo. The details of these new experiments are described below but in short we show that i) CO2/H^+^-inhibited Vgat+ neurons express Sst (new Figure 2A-B), ii) and chemogenetic inhibition of Sst expressing parafacial neurons, but not Pvalb+ or CcK^+^ populations, modulated baseline breathing in a manner consistent with a role of these cells in regulating baseline respiratory activity (new Figure 5 and Figure 5—figure supplement 1). These new results clearly link our transcriptomic analysis of subpopulations of Vgat+ parafacial neurons to cellular CO2/H^+^ sensitivity and respiratory behavior.

In short, we believe that we have address all noted concerns and hope that the editor and reviewers agree the manuscript is ready for publication. Please seen point-by-point responses below for details.

Note that acceptability will have to be reassessed upon submission of the revised version.Reviewer #1:1. The Results section starts with single cell RNA sequencing data of RTN/parafacial neurons, identifying clusters of glutamatergic and GABA/glycinergic neurons. In itself, this is very valuable data, which could justify a stand-alone paper with further analysis. However, starting the Results section with these data is a bit awkward, as they are poorly linked to the rest of the study. Indeed, the authors provide new subclasses of inhibitory neurons, but then focus the rest of the study on overall inhibitory neurons without attempting to link their functional results with the subclasses of neurons molecularly identified. The transition line 164, to show that TdT is expressed in overall inhibitory neurons, without using tools for further specificity as one would expect after reading the previous paragraph on scRNAseq, is somewhat frustrating.The paragraph starting line 172 shows that a subset of parafacial inhibitory neurons are CO2/H^+^ sensitive, and finishes with a statement (lines 188-190) that the identity of this subset of neurons remains to be determined. Testing whether this subset corresponds to a specific cluster of inhibitory neurons identified in the scRNAseq data presented just before would seem the obvious next step for the present study. The authors could have done immunohistochemistry on the neurons patched, as several of the clusters identified have markers for which antibodies exist, or better (but harder) they could have aspirated the patched cytoplasm and performed qPCR to reach the same results. Such data would not only provide a natural link between the first two experimental sets performed, but also provide more detailed information on the inhibitory parafacial neurons that are CO2/H^+^ sensitive that would significantly enhance the impact of this study.With these comments in mind, if no further experiment is done for this paper, it might be better to move the scRNAseq data to the end of the Results section, to open the work towards more specificity in future studies. On that line, the end of the sentence lines 168-170, which closes the scRNAseq data presentation, is more suited for the end of the Results section, rather than the beginning.

We agree and have invested considerable time and effort into addressing this shortcoming. To identify CO2/H^+^ sensitive Vgat+ neurons, we used slice-patch electrophysiology to characterize Vgat+ parafacial neurons as CO2/H^+^-inhibited or -insensitive based on their firing response to 10% CO2. Once the cell type of interest was identified, we gained whole cell access to fill the cell with lucifer yellow (included in the pipette solution) followed by cell type specific immunolabeling. We found that 5 of 5 CO2/H^+^-inhibited neurons were Sst+, whereas 0 of 5 CO2/H^+^-insensitive neurons were Sst+. These new results (illustrated as new Figures 2Aii, Bii) suggest that Sst+ inhibitory neurons are CO2/H^+^ inhibited.

Next, we used chemogenetics to systematically suppress the activity of each main inhibitory neural population in the parafacial region. Specifically, we made bilateral injections of AAV2-hSyn-DIO-hM4D(Gi)-mCherry into the medial parafacial region of Sst-Cre (JAX #: 013044), Pval-Cre (JAX #: 008069) or Cck-Cre (JAX #: 012706) mice. After two weeks recovery, we found that systemic administration of clozapine increased baseline breathing (room air and 100% O2) in Sst-Cre mice but not the Pval- or Cck-Cre lines. In fact, inhibition of just Sst+ parafacial neurons recapitulated our original observation using a Vgat-cre line (Figure 5—figure supplement 1), suggesting that of these populations of parafacial inhibitory neurons only the SST+ population contribute to respiratory activity. Note that all chemogenetic experiments were compared pairwise to saline control, and we confirmed that clozapine had negligible effect on breathing control animals that did not receive AAV injections (Figure 5—figure supplement 2).

Our RNAseq results show that Sst+ defines the largest inhibitory population that varies based on expression of calretinin (*Calb2*, *Slc32a1*), reelin (*Reln*, *Slc32a1*) and neuronal nitric oxide synthase 1 (*Nos1*, *Slc32a1*). We have not yet determined whether CO2/H^+^-sensitivity is specific to one subset of Sst+ neurons or is a common feature of all Sst+ neurons in this region. These new results are shown in a new figure 5 and the text has been modified accordingly. Note that we opted to keep the scRNAseq results as figure 1 as it sets the stage for identifying populations of CO2/H^+^-sensitive inhibitory neurons and determine their contribution to breathing. We hope the reviewers agrees that these new data make for a more complete and exciting story.

2. In a previous article by the same group (Kuo FS et al., eLife 2019, DOI: https://doi.org/10.7554/eLife.43387), mice expressing a Dravet syndrome-associated Scn1a missense mutation conditionally in inhibitory neurons presented breathing alterations including diminished chemosensitivity, with hypo-excitable inhibitory neurons and hyper-excitable excitatory neurons in the RTN. The present study is proposed as Research Advance of this previous article. To strengthen the continuity with the previous study, the authors could have analysed the expression of Snc1a in their scRNAseq data, to identify whether Scn1a is expressed in a cluster of inhibitory neurons, and link it with the cluster that is responsive to CO2/H^+^. This way, the link would be more evident, and the present study would really make use of the scRNAseq data. In the current state of this study, I somewhat fail to see its strong and direct continuity with the previous work justifying a Research Advance format, it would maybe be better suited as a traditional stand-alone Research Article.

The previous study noted by the reviewer showed that expression of a Dravet syndrome associated ion channel mutation in inhibitory neurons disrupted RTN chemoreceptor function and respiratory activity. These findings underscore the need to understand whether and how inhibitory neurons in the parafacial region contribute to breathing. Here, we meet this need by i) characterizing inhibitory neuron diversity within the parafacial region, ii) identifying which subsets of inhibitory neurons are CO2/H^+^ sensitive, iii) establishing that inhibitory synaptic activity regulate activity of RTN chemoreceptors in a CO2/H^+^ dependent manner, and iv) showing how specific subsets of inhibitory parafacial neurons contribute to respiratory activity. Therefore, we believe these results represent an important extension of our previous study and are entirely appropriate for the Research Advance format.

To further link the present result to our previous study, we agree that it would be useful to characterize expression of Scn1a in the parafacial region. We have included a supplemental figure (Figure 1—figure supplement 3) that shows expression of Scn1a transcript in parafacial cells from 10 day old mice. Consistent with our previous study noted above, we show that Scn1a transcript is widely expressed in the parafacial region including both glutamatergic and inhibitory neurons. We have modified the text to include this point.

3. Was synaptic isolation performed when testing the CO2/H^+^ sensitivity of parafacial inhibitory neurons? Did you also block the usual gliotransmitters? The methods section is not clear, please provide more details, as whether these neurons are intrinsically sensitive to CO2/H^+^ or not is critical for interpretation of these data.

Good point. We performed additional slice-patch experiments to determine whether Vgat+ neurons are intrinsically inhibited by CO2/H^+^. We found that exposure to 10% CO2 decreased activity by 1.7 ± 0.2 Hz under control conditions, 2.6 ± 0.3 Hz when glutamate and GABA/glycine signaling was blocked with CNQX (10 µM), strychnine (2 µM) and gabazine (10 µM), and by 1.1 ± 0.1 Hz when purinergic receptors were blocked with 8PT (10 µM) and PPADS (100 µM). Note that CO2/H^+^ evoked ATP release from astrocytes is the only form of glial communication implicated in RTN chemoreception (PMID: 20647426) and previous work showed that inhibitory input to RTN neurons is augmented by P2X signaling (PMID: 16822980). Therefore, for these experiments we targeted purinergic signaling as a means of probing for roles of glia in this response. These results are shown a new Figure 2—figure supplement 2.

Reviewer #2:[…] 1. In the first paragraph of the discussion, the authors conclude that inhibitory inputs to chemosensing RTN neurons are withdrawn/inhibited during exposure to high CO2. The net result is a lessening of GABA restraint on chemosensing neurons and an increase in their neural drive and neurotransmitter release. Another valid interpretation could be that increased neurotransmission from other tonic inputs to the RTN (arising from chemoreceptors activated during high CO2 exposures) contributes to the ventilatory response? Ie the VGAT inhibitory input isn't withdrawn but rather contributes much less to RTN drive during exposure to high CO2. Please include this interpretation, and other alternatives in the discussion.

As suggested by the reviewer, we think CO2/H^+^ increases excitatory synaptic drive to RTN chemoreceptors while simultaneously decreasing inhibitory synaptic input. To support this, we show that exposure to high CO2/H^+^ increased the frequency of excitatory postsynaptic currents (EPSCs) (Figure 4) and decreased frequency of inhibitory postsynaptic currents (IPSCs) recorded from RTN chemoreceptors (Figure 3). Note that these experiments were performed using a Cs+ based internal to block all postsynaptic K^+^ channels (including TASK2 and leak K^+^ targeted by GPR4). We also study the effects of CO2/H^+^ on EPSCs and IPSCs in relative isolation by holding membrane potential at the reversal potential for chloride (-60 mV) or AMPA receptors (0 mV). For example, to study IPSCs we held membrane potential at 0 mV which eliminated AMPA mediated EPSCs. Under these conditions, the observed effect of CO2/H^+^ on IPSC frequency is not due (at least directly) to changes in excitatory input. However, to the reviewers point, CO2/H^+^ may elicit the release of non-glutamatergic neuromodulators that alter postsynaptic conductance to indirectly effect postsynaptic current amplitude. We cannot exclude involvement of this mechanism in the observed CO2/H^+^-induced decrease in IPSC amplitude. Therefore, we have added the following statement to the discussion “It should also be noted that CO2/H^+^ may elicit release of non-glutamatergic neuromodulators that alter postsynaptic conductance and potentially contribute to diminished IPSC amplitude; however, we consider this unlikely since such a non-specific mechanism is expected to have similar effects on both excitatory and inhibitory synaptic currents”.

2. DREADD-mediated inhibition of inhibitory RTN neurons increased baseline minute ventilation but had no effect on ventilatory responses to CO2. Figure 6D-E – most columns do not include data from n=11 mice. Please explain why certain mice/replicates were excluded from the dataset. Figure 6 figure legend is incomplete – there is no description of the data shown for control Vgatcre (Figure 6F-H). Also, there are two Figure 6F.

Thank you for your comment. Results from all 11 mice are shown in Figure 6C-I (now moved to supplemental figure 1). Some data points overlap making is somewhat difficult to see all individual data points. We have restructured this figure (now Figure 5—figure supplement 1) to omit any panel numbering redundancies.

3. CO2/H^+^ synaptic properties of RTN neurons studies (page 8, line 193):"As previously defined (26), RTN neurons were considered chemosensitive if they show some level of spontaneous activity under control conditions and a robust firing rate response to 10% CO2 (1.6 {plus minus} 0.36 Hz; N=10)." – 10% CO2 FR response of 1.6 {plus minus} 0.36 Hz does not seem 'robust' when compared to other previous studies. For example, Wang S et al. 2013 Figure 2D (citation #64), demonstrates FR at pH 7 (~8% CO2) to be >4 Hz for both TypeI and TypeII dissociated RTN neurons. Wouldn't you expect FR to be >>4Hz in 10% CO2, for RTN neurons in an intact brain slice?

The CO2/H^+^ response of RTN chemoreceptors described here is consistent with what has been previously described for type I RTN chemoreceptors in both the brain slice and acute dissociated preparations. For example, we show here that increasing CO2 from 5% (pH 7.3) to 10% (pH 7.0) increased activity by 1.6+/- 0.36 Hz. The study by Wang S et al., noted by the reviewer showed that acutely dissociated RTN chemoreceptors fire at ~2 Hz under control conditions (pH 7.3) and increase their firing discharge to ~3.5 Hz in pH 7.0. Thus, 0.3 pH unit acidification increased activity by ~ 1.5 Hz. Likewise, RTN chemoreceptors in medullary slices showed a similar degree of CO2/H^+^ sensitivity (PMID: 15558061).

4. Figure 2 and Figure 2 legend:Figure 2B – error bars appear to be missing for this n=4 quantified data.

This mistake has been corrected.

Figure 2Aii is an enlarged version of the image at bregma -6.05mm. You could omit this, I don't think it adds anything to the paper. Alternatively, magnify further, use arrows to draw attention to cells of interest.

We agree. The figure panel has been omitted.

Figure 2 legend: include abbreviations (7N, b).

Added.

5. Figure 3C legend: It is not clear what the summary data corresponds to. Indicate this in the figure legend as follows: ' firing responses of CO2/H^+^ -inhibited (n=20, blue), -insensitive (n=53, grey) and -activated (n=7, red). What parameters do the authors use, to define inhibited versus activated neurons?

All VGAT+ cells that responded reversibly to 10% CO2 with ≥ 20% change in firing were considered CO2/H^+^-sensitive. We have edited the text and figure to make this clear.

6. Figure 5 – please confirm that there are no outliers in these values. One replicate seems to have a much higher baseline sEPSC frequency than the others.

Thanks for bringing this to our attention, outlier analysis was used to identify two cells that were subsequently omitted from the sEPSC dataset, making the total N=8 cells. We have also noted this in the figure legend and transparent reporting forms.

7. Supplementary Figure 1 figure legend states that the colors of distributions correspond to classes shown in Figure 1A however Figure 1A displays only 3 colours, whereas Supp Figure 1D displays 5 colours/classes. Please clarify.

Figure 1—figure supplement 1 shows cut offs for denoting Slc17a6 and Slc32a1 cells. There are only three colors shown- blue, green, and red correspond with Slc32a1, Slc17a6, and Chat, respectively. The top expression graph shows low Slc32a1 and Chat in Slc17a6 cells (3 colors). The bottom expression graph shows low Slc17a6 and Chat in Slc32a1 cells; however, those transcript counts overlap (red and green) making it appear as additional color. We have modified the legend to make this more clear.

8. Authors used RNA seq to identify differentially expressed and coregulated genes in RTN neurons in order to infer biological meaning for further studies on this preeminent central chemoreceptor population. Single cell isolation method and RNA-seq methodology was extremely detailed, valuable information regarding inclusion and exclusion criteria were included. The authors use tSNE to investigate segmentation, clustering of genes in the pFRG transcriptome. The data are convincing, since it is well established using other methods, that ~50% of glutamatergic RTN neurons express galanin. Could the authors clarify how they minimised batch effect, which can occur during the experiment, the RNA library preparation, or the sequencing run?

During library preparation, prep kits from the same batch/lot were used and libraries were constructed in parallel. To limit bias from sequencing on different flow cell lanes, and to remove unwanted biological variation, the scRNA-seq data analysis attempted to minimize batch effects in several ways.

After quality control (removing low quality cells) and data normalization, the subsequent analysis is guided by the selection of a set of influential genes. Often called highly-variable genes, these genes are those with the largest variance in expression relative to their mean expression (dispersion) within different bins of mean expression. Due to their high variance, using these genes as the primary features for downstream analysis preserves the variability in the dataset while greatly reduces its dimensionality. From this set of highly-variable genes, we exclude genes which may represent sources of unwanted biological and technical variation so that they do not influence downstream analysis; mitochondrial, ribosomal, hemoglobin, cell-cycle, and stress-related genes are excluded as are sex-related genes such as Xist and genes on the Y-chromosome.

One of the downstream analysis steps involves computing a nearest-neighbor graph- which is a weighted network connecting neighborhoods of related cells together based on shared (influential) gene expression levels. It’s at this step where we utilize the fast and efficient batch correction tool BBKNN (10.1093/bioinformatics/btz625) which computes a batch-corrected nearest-neighbor graph. This graph is then used for downstream clustering and the ensuing cell-type identification and sub-clustering analysis.

There are 20 global clusters (from 2000 genes) – the distribution of cells/cluster and genes/cluster is illustrated in Supp Figure 1C. The basis for the clustering was not clear – could the authors clarify this and make it clear in the text. It appears (from Supp Figure 1 B) that each cluster is defined by related genes that display a high level of expression within the cluster.

Yes, that is roughly correct. As described above, the analysis is guided by the selection of roughly 2000 influential genes that are fed into principal component analysis and then the construction of a batch corrected nearest-neighbor graph. The clustering assignments come from the Leiden community detection algorithm (10.1038/s41598-019-41695-z) which operates on a graph to yield near-optimal partitioning/clustering at a specified resolution (effective partition size). In other words, cells are organized into communities similar to a social network based on their shared expression patterns of these 2000 influential genes, and cluster labels are given to groups of cells based on how similar cells within the group are to one another versus cells outside the group.

This clustering process is the current standard in the field and is integrated into the two most popular scRNA-seq analysis toolkits, ScanPy (used in this study) (10.1186/s13059-017-1382-0) and Seurat (10.1038/nbt.4096). Furthermore, the analysis of this data can be found in the GitHub repository (https://github.com/TheJacksonLaboratory/ventral-parafacial-neuron-scrnaseq).

Reviewer #3:RTN is a small cluster of lower brainstem neurons with a well-described molecular signature. These neurons respond vigorously to CO2 in vivo, contribute to arterial PCO2 homeostasis and maintain breathing automaticity during sleep. Their synaptic inputs are not well identified however and the way in which they encode brain PCO2 in vivo is complex and much debated. Several mechanisms have been proposed (intrinsic pH sensitivity of RTN neurons, astrocyte-mediated paracrine effect of acid, paradoxical acid-mediated contraction of microvessels and synaptic inputs from acid-activated neurons, e.g. serotonergic). The present study explores the possible contribution of local inhibitory neurons to the activity and acid-sensitivity of RTN neurons. Consistent with prior EM evidence, the results show that RTN neurons receive GABA/glycinergic inputs. The contribution of a subset of this input to the CO2 response of RTN neurons is suggested by the results of experiments in slices but the existence of this mechanism is not convincingly supported by the in vivo experiments.The authors confirm the transcriptome of RTN neurons and describe that of several subsets of inhibitory neurons (likely GABA- and glycine-ergic) that reside in their vicinity. This work greatly extends existing observations regarding these local inhibitory neurons. Unfortunately this work did not help identify the particular inhibitory interneurons postulated to be CO2-responsive.Next, the authors show that a subset of these inhibitory neurons are mildly inhibited by acidification in slices. The effects are small and variable from cell to cell but suggest that CO2-triggered disinhibition could perhaps contribute to the overall excitatory effect of CO2 on RTN neurons.This hypothesis is then tested, by determining whether chemogenetic inhibition of these inhibitory interneurons enhances breathing and the respiratory chemoreflex in conscious mice. The authors show is that inhibition of the GABA/glycine interneurons (via a Gi-coupled DREAAD) in unanesthetized mice elevates breathing slightly at rest but seems to have no effect on the breathing stimulation elicited by CO2. The increase in resting breathing is consistent with prior evidence that breathing is increased when a GABA-receptor antagonist is microinjected into the region containing the RTN neurons (Nattie et al) but this fact does not demonstrate that the increase in breathing is caused by activation of RTN neurons. As the authors acknowledge in the discussion the increase in breathing produced by inhibiting GABA interneurons in this brain region could also result from arousal. The increase in breathing could also result from the disinhibition of neurons other than RTN, for example the C1 neurons.Finally, the main interpretative difficulty is that inhibition of the GABA/glycine interneurons does not change the respiratory chemoreflex; this is difficult to reconcile with the results obtained in slices and makes one wonder whether the acid inhibition of the GABA interneurons observed in slices is an artifact or at least whether it is a phenomenon that has any relevance to the CO2 response of RTN neurons in vivo.

We thank the reviewer for their constructive suggestions. We hope that we have satisfactorily address noticed concerns in our point by point responses below.

1. Re: results line 177. Although the authors are probably correct, they need to validate the assumption that their reporter line (Vgatcre::TdT) does indeed "allow for selective targeting of inhibitory neurons in the region of interest". If this has been demonstrated before please give the specific quote. If not, the authors must demonstrate that the vast majority of the tomato-red positive neurons in their region of interest express VGat (transcript or protein) at the relevant postnatal age.

Agreed. We confirmed that 81% and 83% of TdT labeling co-colocalize with Gad67 in P7 and P20 *VgatCre::TdT* mice, respectively (Figure 2—figure supplement 1). At both developmental timepoints we found 8% of cells labeled with TdT only and the remaining were Gad67 only. Furthermore, cell type-specific expression of Cre recombinase in Sst-Cre (PMID 24690741), Pvalb-Cre (PMID 31160332) and Cck-Cre (PMID 33173030) lines used here have been confirmed previously. Therefore, we are confident in our ability to selectively target these populations for in vivo functional assessment.

2. Re: lines 240-241. "Specifically, we bilaterally injected AAV2-hSyn-DIO-hM4D(Gi)-mCherry (10 nL/side, Addgene) into the medial portion of the RTN in VGAT-cre mice". The authors need to also demonstrate here that VGat+-neurons were selectively transduced (i.e. that mCherry and VGat were colocalized). The selectivity of this type of approach is typically overrated (ectopic expression of Cre and Cre-independent effects of the DIO viral prep) and must be verified.

Agreed. We confirmed that 99% of AAV mCherry labeled cells were Gad67-immunoreactive. We also confirm that 95% of Gad67-immunoreactive neurons in the parafacial region were co-labeled with mCherry (Figure 5—figure supplement 1). These results confirm the specificity and efficiency of our viral expression system.

3. Re: results lines 177-183. The defining criteria for pH sensitivity is fine but the results are less than impressive: 20% of neurons were inhibited to some degree (-20% or more) when exposed to a fairly severe acidification (pH 7.0), 66% were unaffected and 9 % were judged to be excited. Figure 3C should be replaced with a distribution histogram summarizing the % response of all the VGAt neurons sampled (e.g. 10% effect binning). Small positive or negative effects of acidification on unit-activity in slices have frequently been reported in other brainstem areas such as the nucleus of the solitary tract. Cherry-picking neurons that are excited and those that are inhibited is not terribly convincing unless this grouping correlates with an identifiable biochemical or structural marker, which was not the case here.

We performed additional experiments and found that 48 of 130 (37%) parafacial Vgat+ cells are robustly and reversibly inhibited by 10% CO2. The fact that most inhibitory neurons in this region do not respond to this level of CO2 is reassuring because it suggests that our stimulus is not a sledgehammer. It also suggests those cells that do respond are different and perhaps specialized sense and respond to CO2 and considering this is an important respiratory chemoreceptor region, it’s tempting to speculate these CO2/H^+^-inhibited parafacial neurons contribute to RTN chemoreception. As suggested, we included a histogram showing CO2/H^+^-induced change in firing for all Vgat neurons sampled (Figure 2D).

We have also determined that CO2/H^+^-inhibited parafacial neurons are SST-immunoreactive, whereas Vgat+ neurons that do not respond to CO2/H^+^ are also not SST-immunoreactive (Figure 2A-B). Based on this and guided by our RNAseq results that identify the main subsets of parafacial inhibitory neurons (Figure 1C), we went on to show that chemogenetic inhibition of Sst+ cells (Figure 5A) but not Pvalb+ (Figure 5E) or CcK^+^ (Figure 5F) parafacial neurons increased baseline breathing in manner consistent with disinhibition. These results also mimic the respiratory phenotype elicited by silencing all parafacial inhibitory neurons (Figure 5 Supplement 1), thus suggesting Sst+ parafacial neurons have a unique role in control of breathing. The text has been modified extensively to include these new results.

4. Re: lines 248-249 and discussion. The increase breathing could have many causes and may not have nothing to do with CO2 chemoreception. For example, selective activation of the rostral C1 neurons produces a very strong increase in breathing in unanesthetized mice and these neurons are tonically inhibited by GABA-ergic input (the basis of the baroreflex). The increased breathing noted at baseline (no CO2 added) could also denote arousal from sleep, an effect produced by activation of the C1 neurons, the RTN itself and perhaps other neurons in the area.

We agree and have modified the last paragraph of the discussion to make clear that parafacial inhibitory neurons (Sst+ in particular) likely regulate breathing at multiple levels of the respiratory circuit. It is worth noting that activation of adrenergic C1 has been shown to increase sigh activity, whereas CO_2_ or optogenetic activation of chemosensitive RTN neurons has been shown to increase respiratory output but with no change in sighing (PMID: 25325789). Also chemogenetic inhibition of Vgat-cre or Sst-cre parafacial neurons did not elicit obvious sigh activity. Therefore, we do not think C1 neurons have a dominate role in this response.

On the other hand, a yet unknown population of parafacial Vgat+ neurons influence sleep-wake states and so arousal may contribute to baseline breathing responses caused by inhibition of Vgat+ or Sst+ parafacial neurons. This interesting possibility will require additional experiments that are beyond the scope of this study.

[Editors' note: further revisions were suggested prior to acceptance, as described below.]

We apologize for the delay in providing this review. But two out of the three initial reviewers declined to re-review the paper and we had to find two other ones. Consequently, you will find some new concerns. However, be aware that your responses to previous comments have been judged adequate and that the additional experiments now included definitively improve the paper. We are happy to publish the paper in this form.However, please take the time to read carefully all comments and address those that you choose to address. Basically, they highlight the following potential improvements: A better definition of your hypothesis; clarifying the functional connections between the different sets of data; stating unambiguously whether you consider the neurons of interest as part of the RTN/pFRG or not; providing a better description of the inhibitory neurons investigated in terms of distribution and number; and paying attention to the scales in different figures that might be wrong.

Thank you for your time and constructive feedback! In short, we have clarified our hypothesis, avoided the use of terminology that confused CO2/H^+^-sensitive inhibitory cells with RTN chemoreceptors, and made all suggested edits to the text and figures.

However, for the sake of time we chose not to include a detailed characterization of the distribution of parafacial inhibitory neurons or assessment of RTN to Sst+ parafacial connectivity. We hope you and the reviewers find the revised manuscript accepted for publication in *eLife*.

Reviewer #2:[…] Whilst important to point out the additional experiments required to definitively demonstrate that the inhibitory RNAseq cluster corresponds with the H^+^ inhibited inhibitory neurons in the RTN, Reviewer 1 acknowledges the difficult nature of these experiments. The authors could include this as future directions in the Discussion section.

We thank you for this comment. We have included this as a future direction in the discussion.

Reviewer #4:[…] 1. It would be essential to define the hypothesis of the present study clearly. The authors "considered" the possibility that ventral medullary neurons in the region of the RTN sense changes in CO2/H^+^ and regulate RTN chemoreception by a mechanism involving disinhibition. However, the authors provided evidence for the ventral medullary inhibitory neurons' involvement in neither the RTN chemoreception nor RTN disinhibition. The authors then propose a novel mode of chemotransduction involving regulation of basal breathing by CO2/H^+^-dependent disinhibition. In fact, the experiments involving the chemogenetic inhibition of SSt neurons did not demonstrate that the RTN neuronal function was affected. The neurons in the ventral medullary region close to RTN project to the pre-Bötzinger Complex (PMID: 26855425), and these projections could be involved in the increases of baseline breathing following SSt neurons chemogenetic inhibition.

Good point. We show that chemogenetic inhibition of Sst+ parafacial neurons increased baseline breathing, and our cellular data is consistent with CO_2_/H^+^-dependent disinhibition of RTN chemoreceptors; however, we do not show that CO_2_/H^+^-sensitive Sst+ neurons directly innervate RTN chemoreceptors or that Sst+ parafacial neurons regulate breathing via the RTN. Based on this, we broadly define our hypothesis as “ventral parafacial inhibitory neurons in the region of the RTN sense changes in CO_2_/H^+^ and regulate baseline breathing by a mechanism involving disinhibition”. We also noted in the discussion that it is not yet clear whether CO_2_/H^+^ inhibited parafacial neurons regulate breathing by disinhibition of RTN or other elements of respiratory control including the pre-BötC.

2. There are no clear functional connections between some of the results described. The authors described the presence of a diversity of inhibitory neurons in the ventral medullary region and that the inhibition of SSt neurons increased the baseline breathing without affecting the ventilatory responses to hypercapnia/acidosis. On the other hand, there are data showing that RTN neurons receive synaptic inhibition, which was reduced in response to hypercapnia/acidosis. Therefore, the authors need to demonstrate that: i) the three different populations (activated, insensitive, and inhibited) of SSt neurons are projecting to RTN chemosensitive neurons, and: ii) the chemogenetic inhibition of SSt neurons increases the RTN neurons firing frequency by reducing the inhibitory synaptic transmission. The SSt neurons recorded in vitro were labeled, and the projections to the RTN neurons can be revealed in the same slice. Besides, the effects of SSt neurons inhibition (chemogenetics) on the RTN firing frequency and inhibitory synaptic transmission can also be analyzed in in vitro experiments using the transfected animals. It is important to note that without new experiments to prove the role of the inhibitory ventral medullary SSt neurons in the RTN chemosensitivity, the authors should consider presenting the data in two (I – somatostatin-expressing RTN neurons regulate baseline breathing; II – the effects of hypercapnia/acidosis on the synaptic transmission to RTN neurons) different manuscripts.

We agree that our results fall short of definitively proving that Sst+ parafacial neurons directly regulate activity of RTN chemoreceptors. Specifically, we show that i) Sst+ parafacial neurons are inhibited by CO_2_/H^+^, ii) RTN chemoreceptors receive inhibitory input that is withdrawn during CO_2_/H^+^; iii) bath application of GABA and glycine receptor blockers increase baseline activity of RTN chemoreceptors (new results), and iv) chemogenetic inhibition of Sst+ but not Pvalb+ or CcK^+^ parafacial neurons increased baseline breathing. Based on this, we think it is reasonable to suggest Sst+ parafacial neurons regulate activity of RTN chemoreceptors. We also believe these complementary results should be published as one manuscript.

3. The authors demonstrated that the inhibitory neurons are located close to the RTN chemoreceptors. In fact, it is possible to observe in the provided images that the Tdt cells of Slc32a1 animals are in the RTN. The anatomical nomenclature is highly confusing in the manuscript. There is no evidence that the described inhibitory neurons are outside of the RTN. Parafacial or even pFRG terminologies only add to the confusion regarding the cell types present in this region of the brain and their role. Therefore, the authors studied the inhibitory neurons of the RTN of mice.

We appreciate the reviewers point. All cells included in this study were located in the ventral parafacial region that overlaps with the RTN. However, since the term RTN is typically used to define CO_2_/H^+^ activated glutamatergic neurons in this region (PMID: 31635852) (clusters 1-2 in Figure 1B), we referring to CO_2_/H^+^ sensitive inhibitory neurons as ‘inhibitory neurons of the RTN’ is more confusing then CO_2_/H^+^-inhibited parafacial neurons. We used this terminology throughout the manuscript.

4. Is the CO2/H^+^ sensitivity of RTN neurons different before and after the inhibitory synaptic transmission blockade? In other words, what is the contribution of the synaptic disinhibition to the RTN neuronal firing response to hypercapnia/acidosis? The RTN neurons receive inhibitory synaptic inputs with very low frequency (0.2 Hz) and amplitude (12 pA), and hypercapnia/acidosis reduces it by nearly half. Therefore, in which extension such rare and small events contribute to the increases in RTN neurons firing frequency and breathing after SSt neurons chemogenetic inhibition?

Thank you for bringing up this point. We added new in vitro results to show that bath application of bicuculine (10 µM) and strychnine (2 µM) increased activity of RTN chemoreceptors by 0.7 ± 0.2 Hz (T_12_=2.201, p=0.022). This finding suggests inhibitory input partly limits activity of RTN chemoreceptors under baseline conditions. We also found the firing response to 10% CO_2_ was similar under control conditions and in the presence of bicuculine and strychnine (Δ -0.3 ± 0.2 Hz; T_12_=1.246, p>0.05). These results are consistent with in vivo chemogenetic experiments showing that inhibition of parafacial Sst+ parafacial neurons increased baseline activity but with no change in the ventilatory response to CO_2_. We have added these results to the text.

5. The authors should provide the number and the rostro-caudal distributions of transfected SSt, CCK and Pvalb neurons, not only the "center" of bilateral virus injections or percentage of neurons, in the chemogenetic experiments. It is an important issue to report considering data reproducibility by other groups, and the extension of the transfected cells in the ventral medulla can provide evidence for different clusters of inhibitory neurons in mice controlling different functions.

We have not systematically characterized the rostro-caudal distribution of transfected cells for each cre line. However, as shown in Figure 5A—figure supplement 1, injections were centered in the RTN region and diffused ~600 µm in the rostro-caudal direction and spread laterally to include expiratory parafacial neurons

6. What about the anatomical distribution of insensitive, activated, and inhibited SSt neurons in the RTN? The authors could provide this information in Figure 2.

CO_2_/H^+^-insensitive and -inhibited neurons were found in both the medial and lateral parafacial regions. Unfortunately, only a limited number of recorded cells were filled with a fluorescent dye so we are not able to provide more information regarding locations of these cell types at this time.

Are the capacitance values different between these neurons? In this regard, the neurons seem bigger (~100 µm) than other neurons described in the RTN based on the scales provided in Figure 2, panels Aii and Bii. Are the scales right?

Thank you for this point. The scale bars in Figure 2 Aii and Bii should be 50 μm not 100 um. The figure has been corrected.

7. Please remove the word "tonic" during the description/discussion of RTN neurons' synaptic inhibition. The authors did not evaluate whether or not the inhibitory synaptic events are tonic because there are no respiratory oscillations in the applied in vitro preparation.

Thanks. We no longer use the word ‘tonic’ to refer to inhibitory input to RTN neurons under control conditions.

8. Is the scale for measured current of the Figure 4 panel A right? The grouped data in panel B show that the amplitude of sEPSCs is -10 pA, but the representative traces show that it is the level of the electrical noise. Besides, the VC traces in Figure 3 panel A are not representing the grouped data of sIPSCs amplitude in panel B. The events are ~ 50 pA, while the grouped data are ~12.5 pA.

Thanks. The scale bars in Figure 3A and 4A were incorrectly labeled, 3A should be 25 pA (not 50 pA) and 4A should be 10 pA rather than 20 pA. The figures have been corrected.

9. Please check the Cl^-^ reversal potential; it is unlikely to be – 60 mV using the described intracellular and extracellular solutions.

Thank you for bringing this to our attention. The reviewer is correct, the Cl^-^ reversal potential for our experimental conditions is approximately -84 mV. Therefore, the holding potential used for recording EPSCs was depolarized to the Cl^-^ reversal potential. However, we consider this a minor issue because potential contaminating IPSCs could easily be excluded from EPSC analysis based on the direction of synaptic currents; inward for EPSCs and outward for IPSCs. We also pharmacologically confirmed EPSCs as glutamatergic at the end of each experiment by bath application of CNQX. We have clarified these points in the text.

10. Discussion section, line 330: Expiratory neurons in the lateral parafacial region do not express Phox2b in rats (PMID: 28004411). Besides, what evidence shows that the expiratory neurons in the lateral parafacial region are not sensitive to hypercapnia/acidosis?

It is not entirely clear whether (or not) expiratory neurons in the lateral parafacial region express Phox2b. The study noted by the reviewer (PMID: 28004411) showed in rats that expiratory parafacial neurons are not Phox2b-immunoreactive. They also showed that blocking glutamate receptors in the lateral parafacial region minimally effected CO_2_/H^+^-induced active expiration. These results suggest expiratory parafacial neurons are distinct from and not dependent on RTN chemoreceptors. Conversely, other work in rats showed selective activation (PMID: 32973046) or inhibition (PMID: 20844141) of ventral parafacial Phox2b+ neurons increased and decreased expiratory activity, respectively. These results suggest one population of Phox2b+ parafacial neurons regulate both inspiratory and expiratory activity. We have clarified this in the text.

It is also possible expiratory parafacial neurons are CO_2_/H^+^-sensitive so we have modified the text to make this clearer.

11. Discussion section, line 413: There is no evidence that disinhibition of Phox2b neurons in the lateral parafacial region evokes forced expiration in rats. In fact, previous studies (references: 19 and 53 and PMID: 28004411) demonstrated that bilateral disinhibition of neurons of the lateral parafacial region, which are located more laterally to the Phox2b positive RTN neurons, induces forced expiration in rats.

Sorry for this confusion. We now refer to these cells as lateral parafacial neurons.

Reviewer #5:[…] Although I have no major concern about this paper, please see below for my comments.1. Page 9, line 212-214, "We found that 5 of 5 CO2/H^+^ -inhibited cells were Sst-immunoreactive (Figure 2Aii) and were not immunoreactive for Pvalb or Cck (data not shown), whereas 0 of 5 CO2/H^+^ -insensitive cells expressed Sst (Figure 2Bii).": This part is most important parts because it was the results indicating that Sst neurons were indeed inhibited by hypercapnic stimulation. Before this part, they showed 48 of 130 (37%) cells that were thought to be inhibitory neurons were inhibited hypercapnic stimulation (page 8). To further clarify properties of these cells, authors investigated 5 cells with combination of whole-cell recordings and immunoreactive examination and showed that cells that were inhibited by hypercapnic stimulation were Sst-immunoreactive. However, this result did not simply mean that all of above 37% neurons were Sst-immunoreactive. Although in vivo experiments supported the idea, is this number (n=5) really enough for the verification? In addition, I suppose that examination for Pvalb or Cck immunoreactivity was performed in cells different from the Sst examination group. Please describe the number of cells tested for Pvalb and Cck. Relating this issue, it would be important to show an example of detailed distribution of Sst (and maybe Pvalb and Cck) immunoreactive cells in the parafacial region together with Phox2b-expressing cells.

We recognize your concern. However, please note that in addition to showing that 5 of 5 CO_2_/H^+^ inhibited cells were Sst+ we also showed that 0 of 5 CO_2_/H^+^ insensitive cells were Sst+. Based on this result we tentatively ruled out involvement of Pvalb and CCK neurons in CO_2_ sensing, and thus did not expand our IHC assessment of CO_2_/H^+^ sensitive cells to include markers of Pvalb or CCK. This decision was further justified by chemogenetic experiments showing selective inhibition of Sst+ neurons but not Pvalb+ or CcK^+^ neurons in the parafacial region disrupted baseline breathing.

Considerable time and effort will be required to characterize the distribution of Sst+ or other subsets of inhibitory neurons in this region. Therefore, to avoid further delays we chose to publish this work without that analysis.

2. Page 8, the response to 10% CO2 was retained when purinergic signaling and fast neurotransmission was blocked: I suppose that these experiments were performed separately but not in the presence of purinergic signaling blockers plus fast neurotransmission blockers. I concern that an uncertainty may remain to conclude that parafacial inhibitory neurons are intrinsically CO2/H^+^ -sensitive.

To confirm, these experiments were performed separately and so it remains possible that synaptic or paracrine mechanisms contribute to CO_2_/H^+^ sensing by inhibitory parafacial neurons. However, preliminary whole cell voltage clamp experiments (in the presence of TTX) suggest CO_2_/H^+^ inhibits Sst+ parafacial neurons directly by activation of an inward rectifying K^+^ conductance. These experiments are ongoing and will be explored in detail in future publications.

3. Page 10, frequency of sIPSCs: "0.22Hz" seems to be rather low. How such low frequency could affect activity of the RTN chemoreceptor neurons?

Good point. To address this concern, we included new data showing bath application of bicuculine (10 µM) and strychnine (2 µM) increased baseline activity of RTN chemoreceptors by 0.7 ± 0.2 Hz (T_12_=2.201, p=0.022). This finding suggests inhibitory input partly limits activity of RTN chemoreceptors under baseline conditions. We also found the firing response to 10% CO_2_ was similar under control conditions and in the presence of bicuculine and strychnine (Δ -0.3 ± 0.2 Hz; T_12_=1.246, p>0.05). These results are consistent with in vivo chemogenetic experiments showing that inhibition of parafacial Sst+ parafacial neurons increased baseline activity but with no change in the ventilatory response to CO_2_. We have added these results to the text.